# Attentional modulation of secondary somatosensory and visual thalamus of mice

Gordon H Petty[1,2], Randy M Bruno[1,2]*

[1]Department of Neuroscience, Columbia University, New York, United States; [2]Department of Physiology, Anatomy, & Genetics, University of Oxford, Oxford, United Kingdom

## eLife Assessment

This study provides an **important** re-evaluation of modality-specific information processing in the thalamus of trained mice. Using an elegant task design that probes competing tactile and visual stimuli, the authors present **compelling** evidence that behavioral training reshapes the sensitivity of higher-order thalamic nuclei. Despite the powerful task design and the significance of the main findings, the origin of the cross-modal responses remains an open question and requires future investigation.

*For correspondence:
randy.bruno@dpag.ox.ac.uk

**Competing interest:** The authors declare that no competing interests exist.

**Abstract** Each sensory modality has its own primary and secondary thalamic nuclei. While the primary thalamic nuclei are well understood to relay sensory information from the periphery to the cortex, the role of secondary sensory nuclei is elusive. We trained head-fixed mice to attend to one sensory modality while ignoring a second modality, namely to attend to touch and ignore vision, or vice versa. Arrays were used to record simultaneously from the secondary somatosensory thalamus (POm) and secondary visual thalamus (LP). In mice trained to respond to tactile stimuli and ignore visual stimuli, POm was robustly activated by touch and largely unresponsive to visual stimuli. A different pattern was observed when mice were trained to respond to visual stimuli and ignore touch, with POm now more robustly activated during visual trials. This POm activity was not explained by differences in movements (i.e. whisking, licking, pupil dilation) resulting from the two tasks. Post hoc histological reconstruction of array tracks through POm revealed that subregions varied in their degree of plasticity. LP exhibited similar phenomena. We conclude that behavioral training reshapes activity in secondary thalamic nuclei. Secondary nuclei respond to the same behaviorally relevant, reward-predicting stimuli regardless of stimulus modality.

## Introduction

Each of the primary sensory cortices are reciprocally connected with corresponding sensory thalamic nuclei. These nuclei are broadly classified as primary or secondary (also known as high-order) (*Guillery and Sherman, 2002*; *Herkenham, 1980*; *Phillips et al., 2019*). The primary nuclei respond robustly to sensory stimulation and transmit sensory information to the cortex (*Chiaia et al., 1991*; *Constantinople and Bruno, 2013*; *Sherman and Guillery, 2002*; *Wimmer et al., 2010*). By contrast, the secondary nuclei receive sparse input from the sensory periphery and are connected with many cortical and subcortical regions. They have nonspecific or complex receptive fields, and their role in sensation and cognition is not well understood.

The rodent primary somatosensory cortex (S1) is connected to the POm, the secondary somatosensory thalamic nucleus (*Deschênes et al., 2016*; *Petersen, 2007*). Despite reciprocal connections with the whisker-sensitive region of S1 (barrel cortex), POm poorly encodes whisker touch and movement (*Diamond et al., 1992*; *Moore et al., 2015*; *Urbain et al., 2015*; *Petty et al., 2021*). However, POm contributes to whisker-dependent behaviors (*La Terra et al., 2022*; *Qi et al., 2022*), facilitates cortical plasticity (*Audette et al., 2019*; *Gambino et al., 2014*; *Williams and Holtmaat, 2019*), and encodes non-sensory task-related behaviors (*El-Boustani et al., 2020*). Despite these recent findings, POm activity in awake, behaving animals is relatively poorly understood.

There is evidence that the secondary nuclei also play a critical role in attention. The lateral posterior nucleus (LP) is the secondary visual thalamic nucleus in rodents, homologous to the lateral pulvinar in primates. In humans, damage to the lateral pulvinar causes attentional deficits (*Danziger et al., 2004*; *Snow et al., 2009*; *Ward et al., 2002*). Similarly, pulvinar silencing in nonhuman primates induces transient impairment in attention tasks (*Wilke et al., 2010*; *Zhou et al., 2016*). Secondary visual thalamus has not yet been similarly studied in rodents.

Despite belonging to ostensibly distinct sensory systems, there are several similarities in the activity and neuroanatomy between LP and POm: Both regions target primary and high-order sensory cortex in the same layer-specific way (*Wimmer et al., 2010*; *El-Boustani et al., 2020*; *Nakamura et al., 2015*; *Roth et al., 2016*), have broad sensory receptive fields (*Moore et al., 2015*; *Roth et al., 2016*; *Ahissar et al., 2000*; *Allen et al., 2016*), and are strongly dependent on cortical activity rather than afferent sensory input (*Diamond et al., 1992*; *Bennett et al., 2019*; *Leow et al., 2022*; *Liao et al., 2010*). These similarities suggest that the secondary nuclei could play similar roles in sensory processing and attention. If the secondary thalamus played a role in attention, the activity in different nuclei would reflect the degree to which attention was allocated to their corresponding sensory modalities. For instance, visual tasks might more strongly activate LP while tactile tasks might more strongly activate POm.

Here, we investigated how sensory-, arousal-, and movement-evoked activity in LP and POm are shaped by associative learning. We conditioned head-fixed mice to attend to a stimulus of one sensory modality (visual or tactile) and ignore a stimulus of a second modality. We then recorded simultaneously from POm and LP. In mice trained to attend to a tactile stimulus and ignore a visual stimulus, tactile responses dominated throughout POm. In contrast, POm showed widespread visual responses in mice trained to respond to a visual stimulus and ignore touch. Unexpectedly, LP exhibited responses similar to POm's. Thus, behavioral training reshapes activity in the secondary thalamus, inducing responses to behaviorally relevant stimuli across the secondary thalamus regardless of sensory modality.

## Results

### Mice can attend to one sensory modality while ignoring a second modality

We developed a conditioning task in which mice were trained to associate a stimulus of one sensory modality with a reward while ignoring a stimulus of a different modality. Water-restricted mice were head-fixed and presented with a visual stimulus (a drifting grating on a monitor) and a tactile stimulus (an innocuous air puff to the distal ends of the whiskers) (*Figure 1a*). Mice were separated into two cohorts: in the 'tactile' cohort a water reward was given at the offset of the air puff (*Figure 1b and c*, top; n=11 mice); in the 'visual' cohort the water reward was given at the offset of the drifting grating (bottom; n=12 mice).

Initially, we trained mice on a shaping version of the task which followed a traditional trial-based structure (*Figure 1b*). A stimulus of one modality or the other was randomly presented in each trial. Trial-based structures are undesirable for our purposes because the unrewarded stimulus is informative that no reward will occur and may garner the animal's attention. Therefore, after reaching criterion (see Methods), mice were moved to the full version of the task. In the full task, the stimuli were presented at the same mean rates as in the shaping task but were completely uncorrelated with one another (*Figure 1c*). After a stimulus, the time until the next stimulus of the same type was drawn from an exponential distribution plus a linear offset (*Figure 1d*, top). The *hazard rate* of the stimulus (the probability of the stimulus occurring at various times since the previous stimulus) was thus flat for

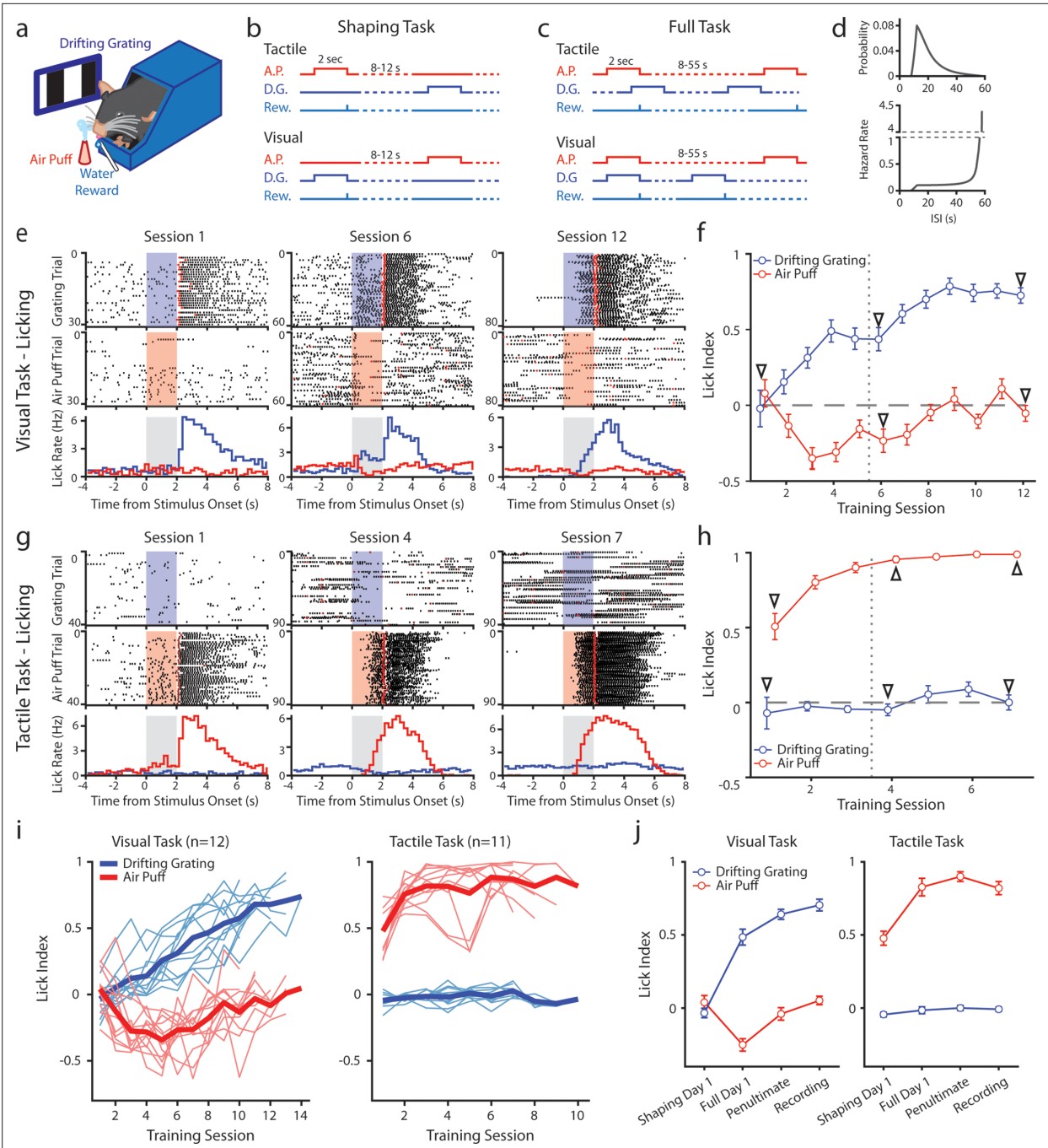

**Figure 1.** Mice can be conditioned to attend to one sensory modality while ignoring a second modality. (**a**) Behavioral setup. Mice were head-fixed and presented with a visual stimulus (a drifting grating on a screen) or a tactile stimulus (an air puff to the distal portion of the whiskers). (**b**) Schematic for the shaping version of the task, depicting the timing of the air puff (red), the drifting grating (dark blue), and the water reward (light blue). Mice were presented with either stimulus with an inter-stimulus interval of 8–12 s. In the tactile cohort, the air puff was paired with a water reward; in the visual cohort, the drifting grating was paired with the water reward. (**c**) Schematic for the full version of the task. Stimuli were allowed to overlap so that only the conditioned stimulus predicted the timing of the water reward. (**d**) Inter-stimulus intervals for each modality in the full task were drawn from an exponential distribution with a linear offset (top). As a result, the hazard rate was largely flat for most stimulus presentations (bottom). (**e**) Example behavior from a mouse trained on the visual task. Top row: raster plot of licks aligned to the onset of the grating (blue region). Red markers indicate the first lick to occur after a reward was available. Middle row: raster plot of licks aligned to the onset of the air puff (red region). Bottom row: Mean lick rate aligned to the onset of the drifting grating (blue) or the air puff (red). Gray region indicates the stimulation period. Data are from the first session of the shaping task (left column), the first day of the full task (middle column), and the final day of conditioning (right column). (**f**) Learning curve of the mouse shown in e, showing the lick index (mean ± SEM) for each stimulus on each training day. Vertical dotted line indicates the day the mouse was

*Figure 1 continued on next page*

*Figure 1 continued*

switched from the shaping task to the full task. Triangles: sessions shown in e. (**g**, **h**) Similar to e and f but for an example mouse trained on the tactile task. (**i**) Learning curves of all visually conditioned mice (left) and tactilely conditioned mice (right). Thin lines: individual mice. Thick lines: mean across all mice. (**j**) Mean and SEM lick index for all mice on the first day of conditioning ('shaping day 1'), the first day of the full task ('full day 1'), the day before neural recordings ('penultimate'), and the day of recording ('recording'). The penultimate day and recording day did not significantly differ in lick index for the conditioned stimulus in either conditioning group (visual conditioning p=0.18, n=12 mice, Wilcoxon signed-rank test; tactile conditioning p=0.08, n=11 mice, Wilcoxon signed-rank test).

The online version of this article includes the following figure supplement(s) for figure 1:

**Figure supplement 1.** Latency of anticipatory licking does not depend on the modality of the conditioned stimulus.

the majority of possible inter-stimulus intervals (ISIs) (*Figure 1d*, bottom). Visual and tactile stimulus intervals were drawn independently, and the stimuli could overlap. As a result, given a stimulus, a mouse could not predict the timing of the next stimulus of either type. In pilot experiments, naive mice trained on the full version of the task learned at a much slower rate than mice initially trained on the shaping version of the task and then advanced to the full task. We, therefore, trained all mice first on the shaping version of the task for at least three days and until they reached the criterion before moving them to the full version (see Methods).

We examined anticipatory licking to assess how mice learned the stimulus-reward association (*Figure 1e–j*). For each stimulus presentation, we computed a lick index (the difference in the number of licks during stimulus presentation and 2 s prior to stimulus onset, divided by the sum of all licks in both periods) (*Jurjut et al., 2017*). A positive lick index indicates that a mouse responded to a stimulus by licking, a negative index indicates that it withheld licking, and an index of zero indicates that it ignored the stimulus.

An example visually conditioned mouse is presented in *Figure 1e and f*. During the shaping task, this mouse initially ignored both stimuli: the mouse did not lick in response to either the air puff or drifting grating, though it did lick to consume the water reward (*Figure 1e*, left). After several shaping sessions, the mouse learned that the drifting grating was predictive of a reward, as evidenced by an increase in licking after the onset of the drifting grating but before reward delivery (*Figure 1e* middle). However, this mouse also learned that the air puff predicted a *lack* of reward, as evidenced by withholding licking upon the onset of the air puff. The mouse thus displayed a positive visual lick index and a negative tactile lick index, suggesting that it attended to both the tactile and visual stimuli (*Figure 1f*, middle arrow).

In the full version of the task, the two stimuli are completely decorrelated, and the air puff is no longer predictive of the presence or absence of reward. After several sessions of conditioning on the full version of the task, the mouse no longer altered its licking in response to the air puff (*Figure 1e*, right; *Figure 1f*, rightmost arrow). This indicates that the mouse learned to ignore the uninformative, distracting stimulus. All visually conditioned mice exhibited a similar learning trajectory (*Figure 1i*, left; 1 j, left). These mice initially learned that the air puff indicated the absence of reward, sometimes before they responded to the visual stimulus at all. They only learned to ignore the air puff after several sessions of conditioning on the full version of the task. By contrast, tactilely conditioned mice rapidly learned the association between air puff and reward, displaying a positive lick index as early as session 1 (*Figure 1g*, left; 1 hr). Tactilely conditioned mice always ignored the drifting grating, never displaying a visual lick index significantly different from zero (*Figure 1i*, right; *Figure 1j* right). Such a discrepancy suggests that mice found this particular tactile stimulus more salient than the drifting grating. All mice were first trained on the shaping version of the task for at least three sessions, and until they learned the association between the conditioned stimulus and reward (Methods). Mice were switched to the full version of the task and trained for at least four more days, and until they no longer responded to the unrewarded stimulus. All tactilely conditioned mice reached the criterion within six to nine sessions, and all visually conditioned mice learned within nine to thirteen sessions.

Though the visual version of the task took longer to learn, there was no difference in how the two groups responded to the conditioned stimulus in fully trained mice: tactilely conditioned mice licked in response to the air puff with the same latency that visually conditioned mice responded to the drifting grating (*Figure 1—figure supplement 1c*, tactile conditioning median latency of 1.1 s, visual conditioning latency of 1.0 s, p=0.98, signed-rank test). In both groups, we observed a broad range of response times across mice, with some mice responding as early as 500 ms after stimulus onset

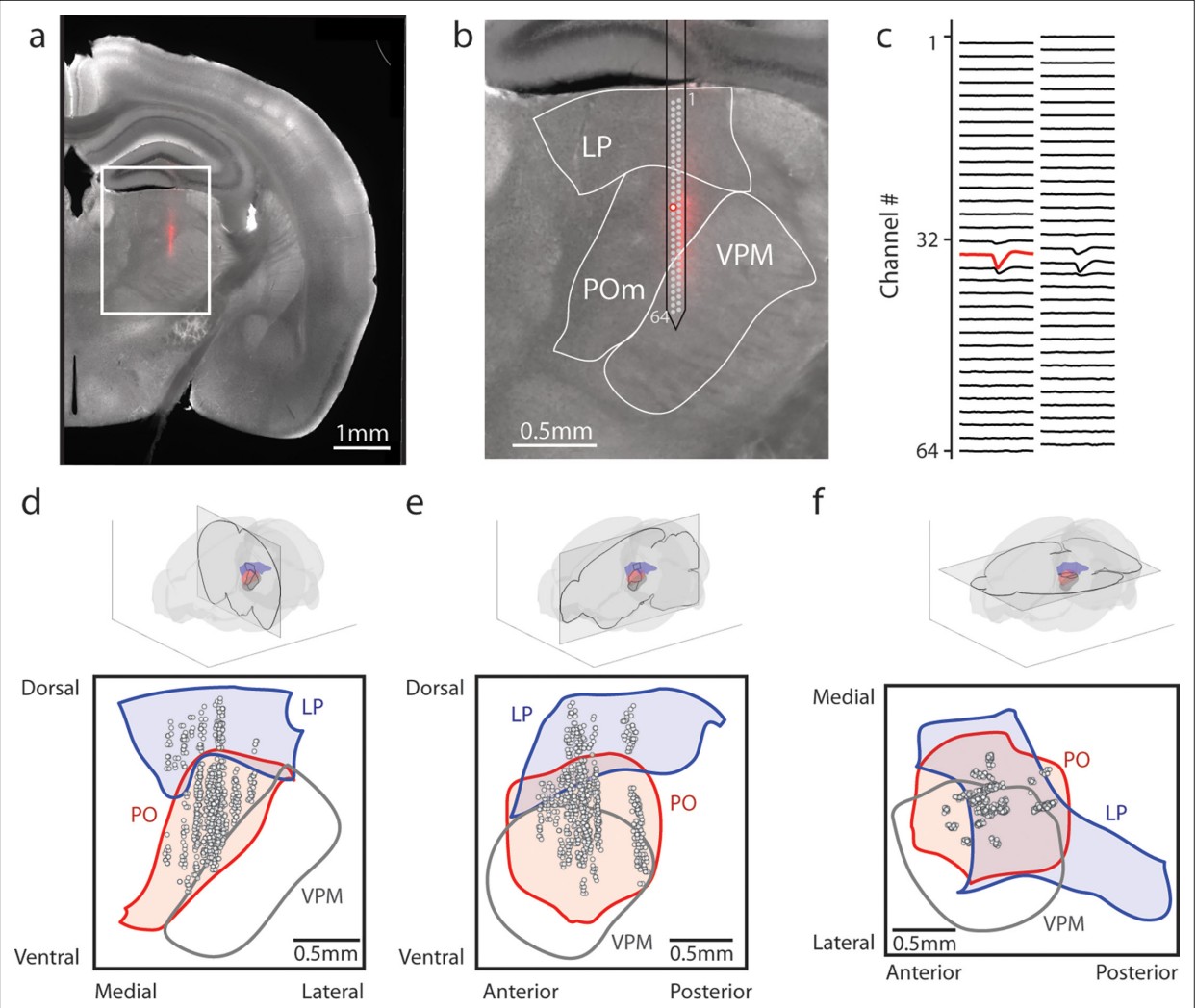

**Figure 2.** Array recordings were made simultaneously from posterior medial nucleus (POm) and lateral posterior nucleus (LP). (**a**) Example coronal section. Red: DiI marking recording track. Gray: endogenous biotin stained with streptavidin Alexa fluor 647. (**b**) Zoom of the outlined region in a. The borders of LP, POm, and ventral posterior medial nucleus (VPM) are outlined. A 64-channel silicon probe was used to record from LP and POm. (**c**) Mean waveform of an example POm cell across all probe channels. The channel with the largest mean waveform was identified and used to estimate the position of the cell (red circle in B). (**d, e, f**) Estimated location of putative cells (gray circles) located in POm (red) and LP (blue) aligned to the Allen reference atlas. Cell locations are jittered along the medial-lateral and anterior-posterior axes for visualization purposes. (**d**) Coronal projection over the range of cell locations in the anterior-posterior axis. (**e**) Sagittal projection over the range of cell locations in the medial-lateral axis. (**f**) Transverse projection over the range of cell locations in the dorsal-ventral axis.

The online version of this article includes the following figure supplement(s) for figure 2:

**Figure supplement 1.** Manual curation of posterior medial nucleus (POm) and lateral posterior nucleus (LP) cells.

and others responding as late as 1.6 s after onset (*Figure 1—figure supplement 1a, b*). In summary, mice can learn to disregard uninformative stimuli, even when those stimuli are in and of themselves highly salient.

## POm responses to tactile and visual stimuli depend on conditioning paradigm

To evaluate the effect of reward conditioning on activity in the secondary sensory thalamus, we performed electrophysiological recordings in mice fully trained on the conditioning task. We inserted a 64-channel multielectrode array that spanned both LP and POm. Probes were coated in DiI, and histology images were aligned (see Methods) to identify the location of each putative cell

(*Figure 2a–c*). Spike clusters were manually curated based on waveform shape and spike autocorrelation (Methods, *Figure 2—figure supplement 1a–d*). We identified 347 POm cells and 64 LP cells from 11 tactilely conditioned mice, and 257 POm cells and 67 LP cells from 12 visually conditioned mice. Our recordings covered most of the volume of POm but were clustered primarily in the anterior and medial portions of LP (*Figure 2d–f*). Cells that were within 50 μm of a region border were excluded from analysis. We observed no difference in waveform shape between POm and LP, though POm cells had on average higher amplitude than LP cells (*Figure 2—figure supplement 1e–i*).

During recording, mice were presented with an equal number of air puffs and drifting gratings (80–120 of each stimulus per mouse, median 100). We first investigated how conditioning affects POm activity. We observed a heterogeneous population of cells with varied patterns of activity. For example, in tactilely conditioned mice, we observed certain cells with a sharp increase in firing rate at air puff onset (*Figure 3a*, left) and other cells that were primarily active after the stimulus offset when the mouse was consuming the water reward (*Figure 3a*, right). A large portion of cells responded most strongly to the stimulus onset, having a peak firing rate within 250 ms of the start of the air puff (70 cells, 20.2%). However, we also observed cells with peak firing rates spanning the entire duration of the air puff and up to 4 s after the air puff offset (*Figure 3c*). Of cells from tactilely conditioned mice, 195 (56.2%) significantly modified their firing rate within 1 s of the air puff onset compared to the baseline firing rate (*Figure 3d*, bottom). Of these cells, only 23 cells (6.6%) had significant changes in firing rate following the onset of the drifting grating (*Figure 3d*, top). No cells responded to the drifting grating alone.

POm cells from visually conditioned mice exhibited a strikingly different pattern of activity (*Figure 3b, e and f*). Surprisingly, many POm cells responded to the drifting grating alone or to both the drifting grating and the air puff (*Figure 3b*, left). Others had no response to either stimulus but were more active during reward consumption (*Figure 3b*, right). Compared to tactilely conditioned mice, a much larger portion of cells responded to the onset of the drifting grating and a smaller portion responded to the air puff (*Figure 3f*, 16 air puff responding cells, 6.2%; 12 grating responding cells, 4.7%; 85 cells responding to both stimuli, 33.1%). At the population level, visually conditioned POm cells had a significantly higher firing rate than tactilely conditioned POm cells during the first second of the drifting grating stimulus period (*Figure 3g and h*). Furthermore, POm visual responses often preceded a mouse's lick response (*Figure 1—figure supplement 1*).

In both conditioning groups, POm activity increased when mice were consuming water. We examined firing rates during the 'offset' period, defined as the 2 s period after stimulus offset and during the reward delivery (or absence of reward for the unconditioned stimulus). In both groups, cells had a greater firing rate during the offset period of the rewarded stimulus than the unrewarded stimulus (*Figure 3h*, Wilcoxon signed-rank test, $p < 10^{-20}$ for both groups).

In summary, POm displayed dramatically different responses to visual and tactile stimuli depending on the task type.

## Conditioning alters POm response latencies to a tactile stimulus

Though POm poorly encodes whisker touch compared to other somatosensory brain regions, whisker stimulation does induce low-latency (<100 ms) responses in certain POm cells, even in anaesthetized animals (*Diamond et al., 1992*). We observed POm cells that responded to air puff onset in both visually and tactilely conditioned mice (*Figure 3c, e and g*). Such responses could be 'purely sensory' (i.e. driven by ascending brainstem inputs) or could result from consequent changes in movement, arousal, and behavioral state.

To investigate how conditioning might affect early tactile stimulus responses, we analyzed the mean firing rates of each POm cell immediately after the air puff at a finer timescale (*Figure 4*). We limited our analysis to the first 500 ms after stimulus onset, as it precedes the majority of licking activity across mice (*Figure 1—figure supplement 1*). In both conditioning groups, we observed a multiphasic population response, characterized by an early peak in firing rate centered at 40 ms post-stimulus (*Figure 4a*). This peak is followed by a prolonged period of elevated firing rates. In tactilely conditioned mice POm activity remained elevated throughout the duration of the 2 s air puff, while in visually conditioned mice POm activity attenuated over air puff presentation (*Figures 4a and 3g*).

As the early peak in population activity was similar in both timing and magnitude across conditioning groups, we considered that there exists a subset of POm cells in both groups with low-latency

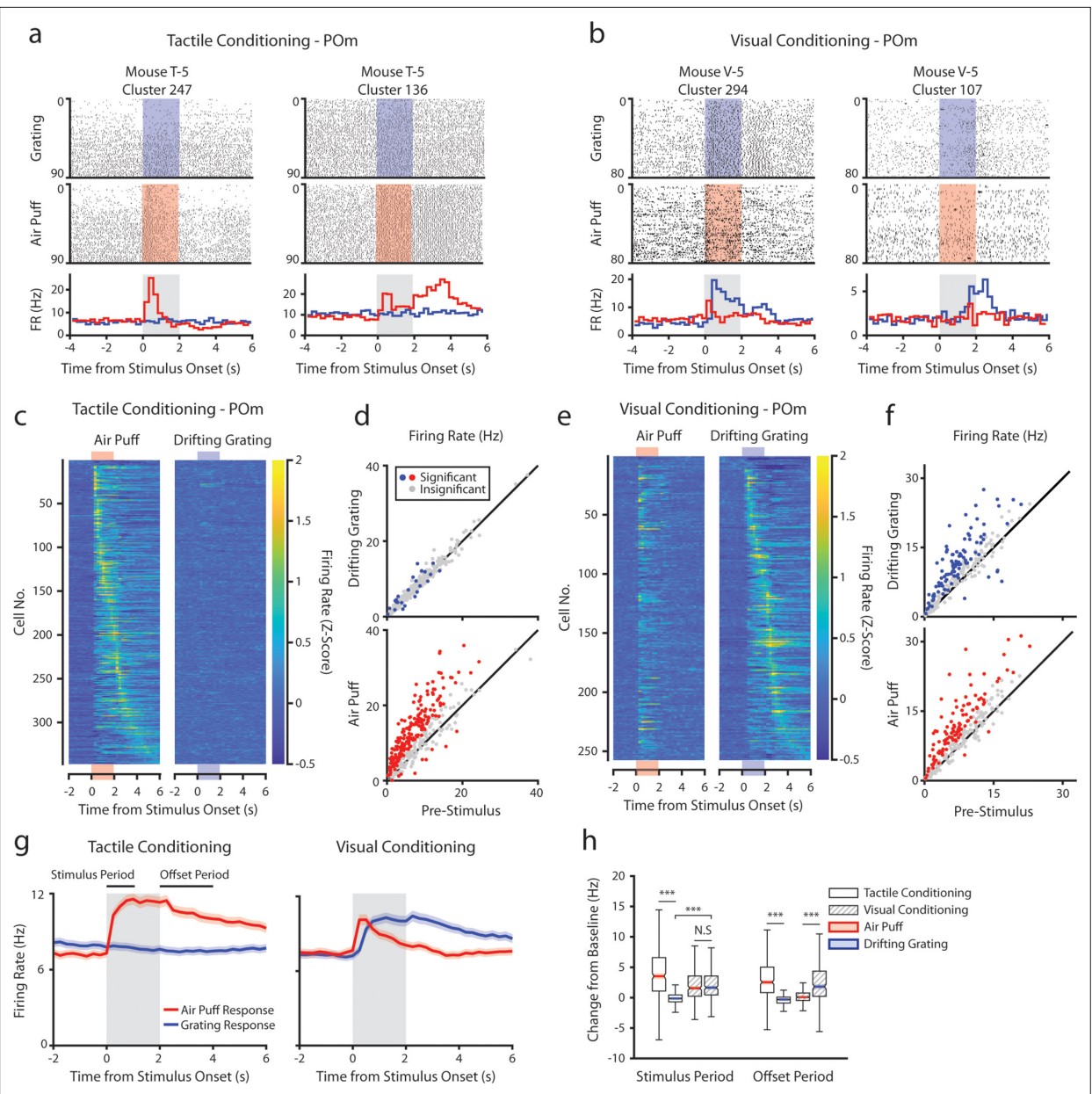

**Figure 3.** Posterior medial nucleus (POm) responses to tactile and visual stimuli depend on the conditioning paradigm. (**a**) Example cells from a tactilely conditioned mouse (same mouse as in **Figure 1g**, session 7). Top row: spike raster aligned to the onset of the drifting grating (blue). Middle row: spike raster aligned to the air puff (red region). Bottom row: Mean firing rate over each trial aligned to the drifting grating (blue line) and the air puff (red line). Gray region indicates the timing of both stimuli. (**b**) Sample cells from a visually conditioned mouse (same mouse as in **Figure 1e**, session 12). (**c**) Firing rate of all POm cells from tactilely conditioned mice aligned to either the air puff (left) or the drifting grating (right). Firing rates are binned at 250 ms and cells are sorted by the timing of the peak firing rate when aligned to the air puff. n=347 cells, 11 mice, range = 11–57 cells/mouse, median = 33 cells/mouse. (**d**) Scatter plot of POm firing rates 1 s pre-stimulus (x-axis) and 1 s during stimulus onset (y-axis) from tactilely conditioned mice. Colored markers: cells with significantly different firing rates pre- and post-stimulus (1-way ANOVA with Holm-Bonferroni correction and post-hoc Wilcoxon signed-rank test). Gray markers: cells without significantly different firing rates. (**e**) Firing rates of all POm cells from visually conditioned mice aligned to either the air puff (left) or the drifting grating (right). Cells are sorted by the timing of the peak firing rate aligned to the grating. (n=257 cells, 11 mice, range=10–42 cells/mouse, median=19 cells/mouse). (**f**) Scatter plot of POm firing rates pre- and post-stimulus onset for visually conditioned mice, as in e. (**g**) Event-triggered average firing rates of all POm cells (mean ± SEM) aligned to either the air puff (red) or the drifting grating (blue) in tactilely conditioned mice (left) and visually conditioned mice (right). Gray region indicates the timing of both stimuli. Rates binned at 250 ms. (**h**) Box plots of the change in firing rate from baseline for each POm cell during the first second of stimulus onset ('stimulus period') and during the first 2 s post-stimulus ('offset period'). In tactilely conditioned mice, the change from baseline was greater in the air puff stimulus period than the visual stimulus period (two-way ANOVA with post-hoc signed-rank test; conditioning type $F$=1.98, p=0.16; stimulus type $F$=153, p<$10^{-33}$; interaction $F$=115, p<$10^{-26}$, signed-rank

*Figure 3 continued on next page*

*Figure 3 continued*

$p<10^{-45}$). There was no difference in the change from baseline between stimulus types in visually conditioned mice (p=0.42, signed-rank test). In visually conditioned mice, the drifting grating stimulus response was significantly greater than that of tactilely conditioned mice (rank-sum test, $p<10^{-42}$). In both conditioning types, firing rates in the offset period were significantly different between stimulus types (two-way ANOVA, conditioning F=0.01, p=0.90; stimulus F=16, $p<10^{-5}$; interaction F=312, $p<10^{-62}$; signed-rank $p<10^{-20}$ for both visually and tactilely conditioned mice).

stimulus responses driven directly by whisker stimulation. For each cell, we computed a stimulus-response latency, defined as the first 10 ms time bin after the air puff when the cell's firing rate differed statistically from the baseline (Methods). Interestingly, we observed a near-identical proportion of cells with air puff response latencies below 100 ms in both conditioning groups (*Figure 4b*; 43.2% of tactilely conditioned cells and 42.8% of visually conditioned cells). Certain POm cells responded as early as 10 ms after the air puff. However, in tactilely conditioned mice nearly twice as many cells responded between 100 and 500 ms after air puff onset (32.0% of tactilely conditioned cells and

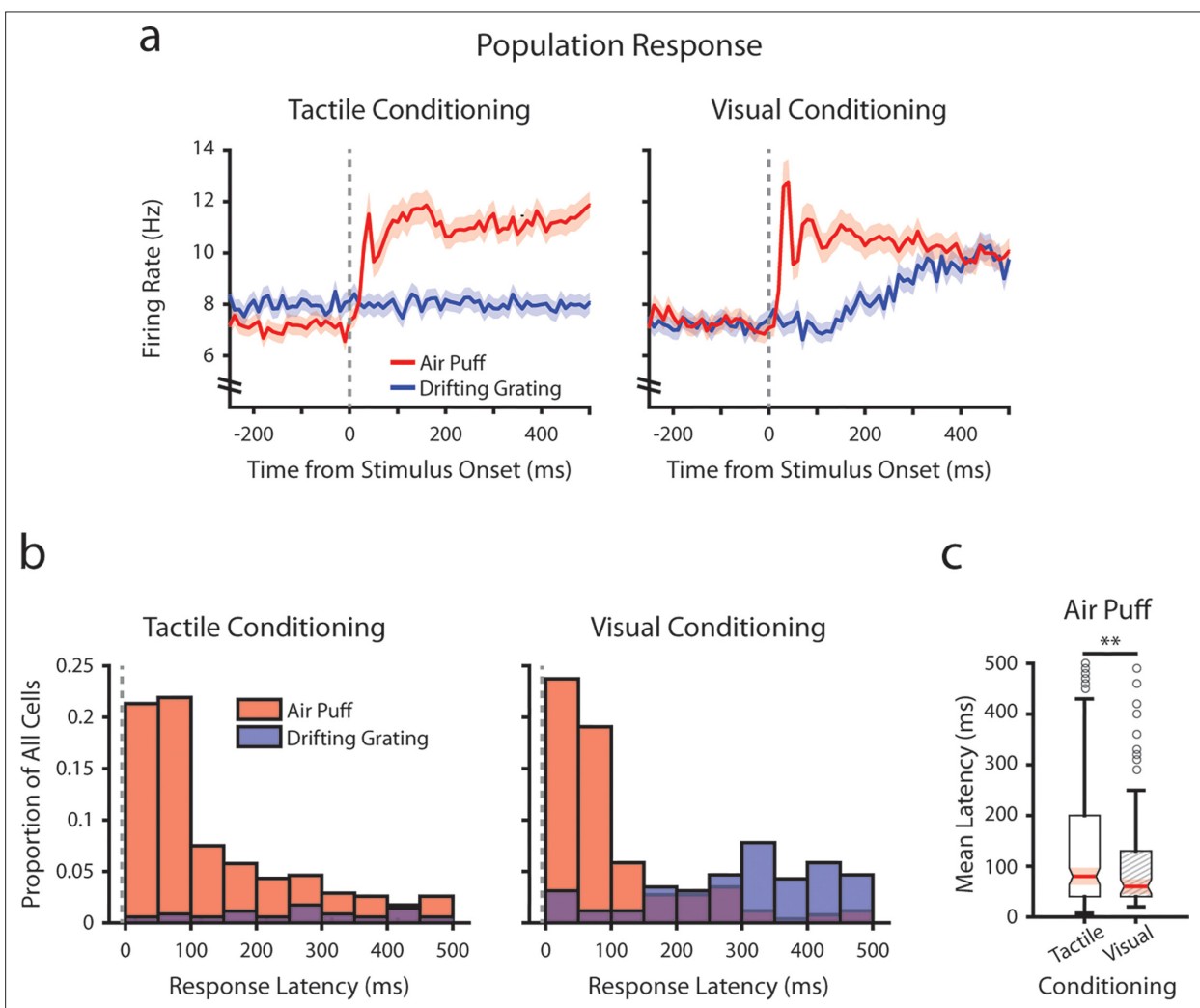

**Figure 4.** Conditioning alters tactile stimulus response latencies in posterior medial nucleus (POm). (**a**) Event-triggered average firing rates of all POm cells (mean ± SEM) aligned to either the air puff (red) or the drifting grating (blue) in tactilely conditioned mice (left) and visually conditioned mice (right), as in *Figure 3g*. Dashed line indicates stimulus onset. Firing rates are binned at 10 ms. Note the truncated y-axis. (**b**) Histogram of the stimulus-response latency, as a portion of total recorded POm cells for each conditioning group. Stimulus-response latency was defined as the first time a cell's firing rate crossed a confidence interval threshold compared to baseline (Methods). Only cells with response latencies less than 500 ms are represented. Red: stimulus-response latency relative to the air puff onset. Blue: response latency relative to the drifting grating onset. (**c**) Box plot of POm early air puff response latencies (≤ 500 ms). Response latencies in tactilely conditioned mice were later than those in visually conditioned mice (p=0.003, Wilcoxon rank-sum test; tactile conditioning median = 80 ms, IQR=[40, 200], n=226 cells; visual conditioning median=60 ms, IQR=[40, 130], n=101 cells).

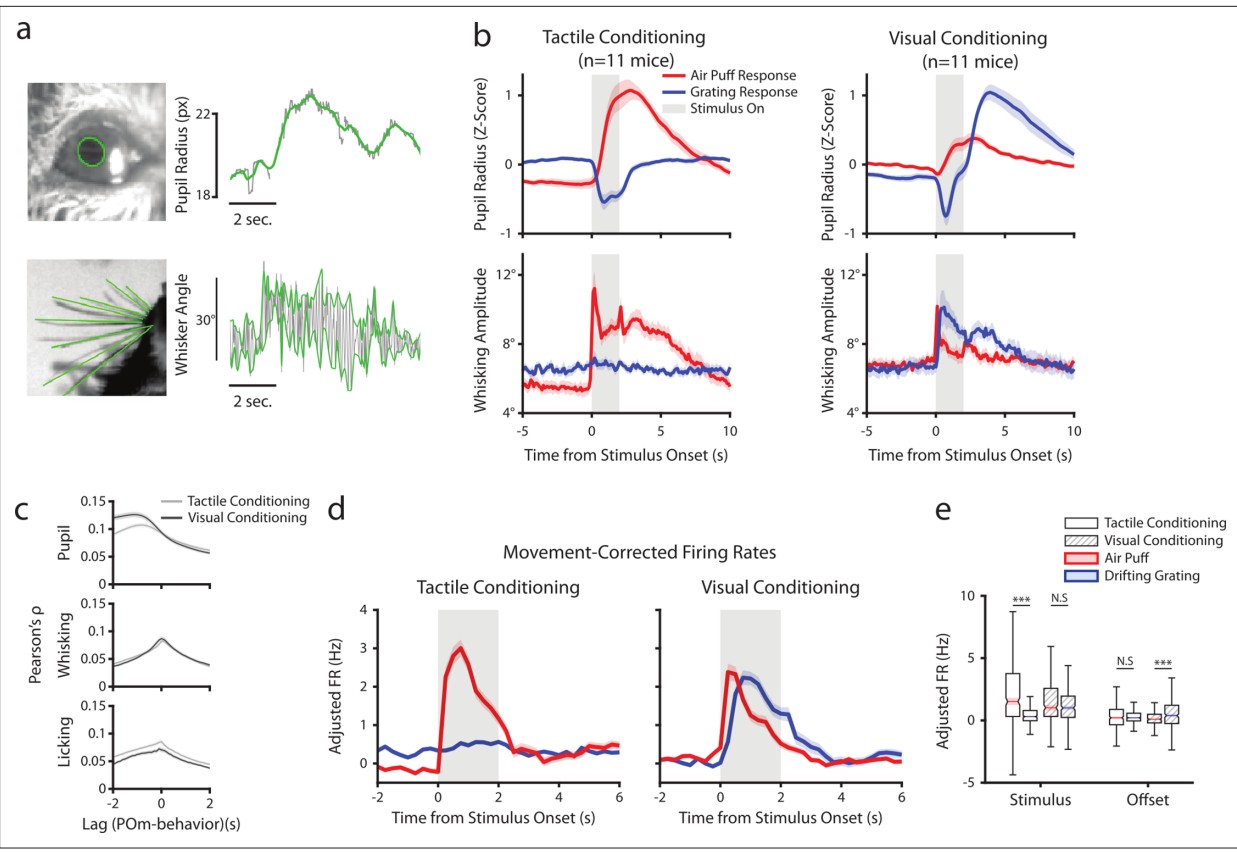

**Figure 5.** Posterior medial nucleus (POm) correlates with arousal and movement regardless of conditioning. (**a**) Pupil and whisker tracking. Top left: Example frame from an eye video with the edge of the pupil highlighted in green. Top Right: A sample trace of pupil radius over time (gray: raw; green: smoothed). Bottom Left: Example frame from a whisker video with whiskers highlighted in green. Bottom Right: Example trace of whisker angle over time (gray) and the most protracted and retracted whisker positions over each whisk cycle (green), which is used to compute whisking amplitude. (**b**) Whisking amplitude and pupil radius aligned to stimuli in tactilely conditioned mice (n=11 mice, left column) and visually conditioned mice (n=11 mice, right column). Top row: Pupil radius of each mouse aligned to either the air puff (blue) or the drifting grating (red) (Z-scored, mean ± SEM). Bottom row: Whisking amplitude of each mouse aligned to the two stimuli (mean ± SEM). Gray regions indicate stimulus timing. (**c**) Cross-correlations between POm activity and pupil radius (top), whisking amplitude (middle), and lick rate (bottom). Light gray: Cells from tactilely conditioned mice (mean ± SEM, n=347 cells, 11 mice). Dark gray: Cells from visually conditioned mice (n=257, 11 mice). Negative lags indicate that POm activity precedes the movement, and positive lags indicate that POm activity follows the movement. (**d**) Movement-corrected firing rates. The firing rate of each cell was fit to a linear model with whisking, licking, pupil radius, and baseline rate as predictors. Model residuals of all cells (mean ± SEM) are aligned to the drifting grating and the air puff, as in B. (**e**) Box plot of the model residuals of each POm cell during the first second of stimulus onset ('stimulus period') and during the first 2 s post-stimulus ('offset period') for the same time windows as in 3g,h. Adjusted firing rates in tactilely conditioned mice were significantly higher during the air puff stimulus period than the drifting grating period. (Two-way ANOVA with post-hoc Wilcoxon signed-rank test. Conditioning type $F=4.23$, $p=0.04$; stimulus type $F=68.2$, $p<0.001$; interaction $F=38.5$, $p<0.001$; tactile conditioning Wilcoxon $p<0.001$; visual conditioning Wilcoxon $p=0.09$). In visually conditioned mice, adjusted firing rates were higher during the drifting grating offset period than the air puff offset period. (conditioning $F=0.003$, $p=0.95$; stimulus $F=3.9$, $p=0.048$; interaction $F=9.45$, $p=0.002$; tactile conditioning Wilcoxon $p=0.94$, visual conditioning $p<0.001$).

18.3% of visually conditioned cells). As a result, the median air puff latency in tactilely conditioned POm cells is later than that of visually conditioned cells (*Figure 4c*, p=0.003, Wilcoxon rank-sum test). Thus, conditioning alters POm activity by changing the proportion of cells that respond later than 100 ms after air puff onset, while the proportion of cells that respond earlier remains constant.

## POm correlates with arousal and movement regardless of conditioning

In freely-whisking mice, POm activity correlates with whisking, pupil radius, and behavioral state (*Moore et al., 2015*; *Urbain et al., 2015*; *Petty et al., 2021*). To assess whether this correlation persists in our conditioning task and whether the conditioning type is a factor, we acquired a video of the whiskers and eye contralateral to the recorded thalamus (*Figure 5a*). We measured pupil radius

as a metric of arousal and quantified whisking activity by *amplitude*, the difference between the most protracted and retracted position of the whiskers on each whisk cycle (Methods). Whisking and pupil responses to stimuli were contingent on task type (*Figure 5b*). Tactilely conditioned mice displayed a robust increase in pupil radius and whisking amplitude at the onset of the air puff (left). This movement and arousal change persisted throughout the offset period. Tactilely conditioned mice did not whisk in response to the drifting grating; however, the grating did induce a pupil constriction as expected from autonomic response to luminance change. We observed a similar pupil constriction at stimulus onset in visually conditioned mice, but this was followed by pupil dilation (right). These mice also displayed increased pupil radius and whisking in response to the unrewarded tactile stimulus, again suggesting that the whisker air puff is inherently more salient than the drifting grating even when a mouse has been trained to ignore it.

Consistent with previous results, we found that POm activity was correlated with both pupil and whisking regardless of conditioning type (*Figure 5c*). Interestingly, POm neurons were slightly more correlated with licking after tactile conditioning than after visual conditioning. We have previously reported that POm correlates with pupil radius most strongly at a time lag of about 1 s (*Petty et al., 2021*). We found that cells were more strongly correlated with pupil radius at this time lag in visual conditioning than in tactile conditioning. This difference in correlation might be due to the between-group difference in pupil dynamics following the drifting grating onset. However, there was no effect of conditioning on the POm-pupil correlation at a time lag of zero.

Much of the stimulus-aligned POm activity observed could conceivably be due to movement and arousal responses rather than afferent sensory signals. To disentangle these, we fit the firing rate of each cell with a linear model using pupil radius, whisking amplitude, and lick rate as predictors on a trial-by-trial basis. We then analyzed the model residuals as a 'movement-corrected' stimulus-evoked firing rate (*Figure 5d and e*). Compared to the raw firing rates (*Figure 3g and h*), we found that regressing out movement had little effect on activity during the stimulus period but resulted in reduced activity in the offset period. Thus, the apparent tactile and visual responses in POm after behavioral conditioning cannot be explained by movement or arousal.

We next sought to determine if this state-dependent modulation was a universal trait of all POm cells, or if only certain cells were coupled to movement and arousal while others responded to stimuli. We again fit a linear model of each cell's firing rate, but this time included the timing of the drifting grating, air puff, and water reward as predictors in addition to the three-movement and arousal variables. Cells with licking, pupil radius, and/or whisking coefficients that were significantly different from 0 were classified as 'movement tuned,' and cells with either of the two stimuli as significant predictors were classified as 'stimulus tuned.' The vast majority of POm cells were tuned to both movement and stimuli: 312/347 (89%) of cells in tactilely conditioned mice and 239/257 (93%) of cells in visually conditioned mice. A subset of cells were tuned only to movement (33/347 (9.5%) tactile, 16/257 (6.2%) visual), and fewer still were tuned only to the air puff or drifting grating (2/347 (0.6%) tactile, 2/257 (0.8%) visual). Movement and arousal relative activity are thus predominant throughout POm, regardless of whether individual cells also possess stimulus-specific responses.

## Modality selectivity varies with anatomical location in POm

We considered the possibility that the differences in stimulus-aligned activity were caused by distinct regions within POm having distinct stimulus responses. As many POm cells responded significantly to both the drifting grating and the air puff (especially after visual conditioning), we computed a stimulus selectivity index to measure a cell's preference for one stimulus over the other. A selectivity index of +1 indicates that a cell responds only to the air puff, an index of –1 indicates that it responds only to the drifting grating, and an index of 0 indicates that it responds to both stimuli with equal magnitude. We limited our analysis to those cells that responded to at least one of the stimuli, having a mean firing rate significantly different from baseline within 1 s of stimulus onset (corresponding to the colored points in *Figure 3d and f*).

All stimulus-responsive cells had a positive selectivity index and were thus tactilely tuned; there were no visually tuned cells in this population (*Figure 6a*). We found no pattern between these cells' anatomical location and their selectivity index (*Figure 6b*, left). In visually conditioned mice, POm cells exhibited a wide variety of selectivity, with some tuned to touch, others to vision, and several responding equally to either stimulus (*Figure 6a*). In these animals, touch selective cells clustered in

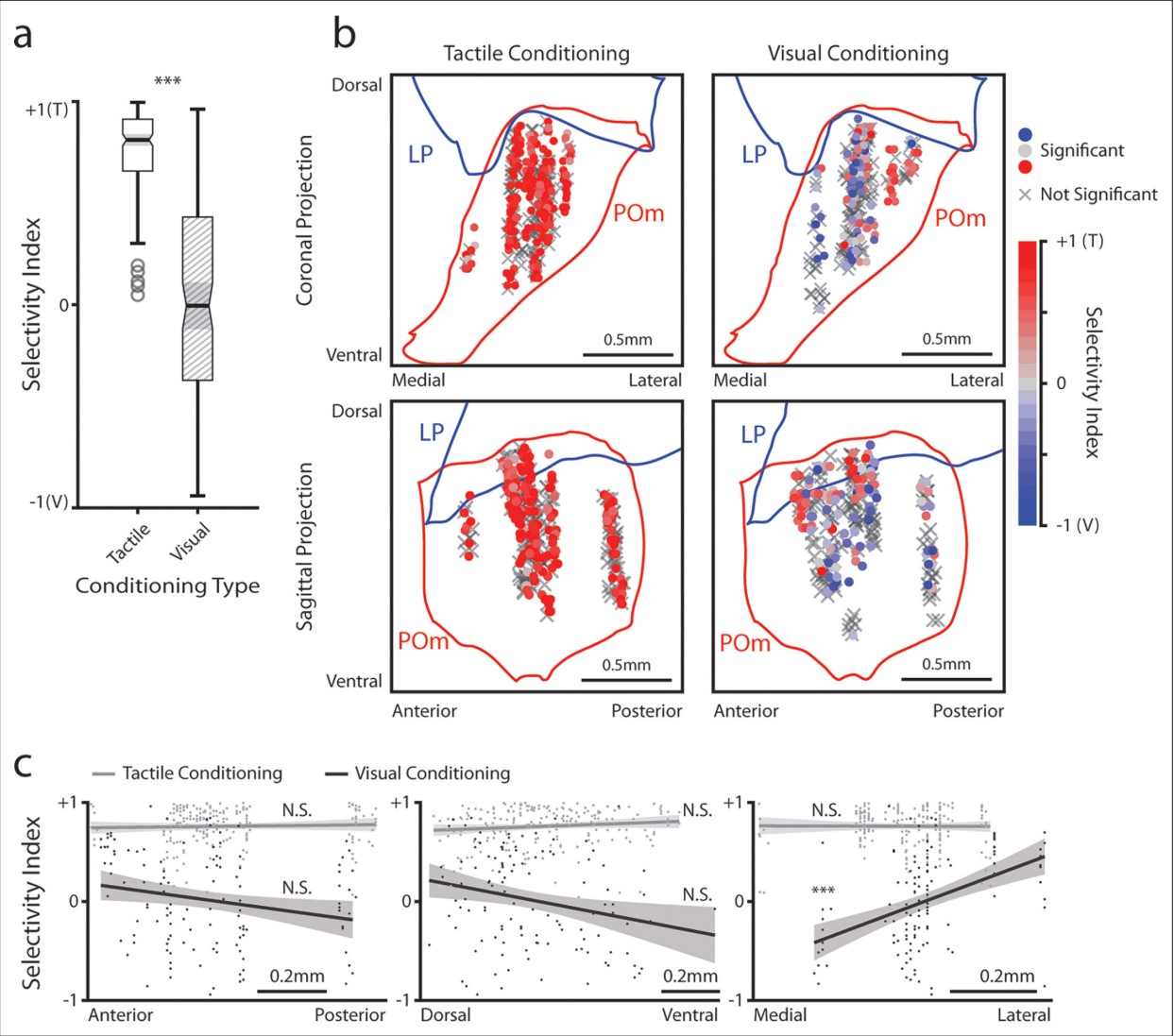

**Figure 6.** Posterior medial nucleus (POm) responses to tactile and visual stimuli vary based on anatomical location. (**a**) Sensory selectivity index for significantly responding POm cells (colored cells in 3d and 3f). Cells with a positive selectivity index more strongly respond to the air puff, and cells with a negative index primarily respond to the drifting grating. Cells from tactilely conditioned mice had a significantly higher selectivity index (Wilcoxon rank sum test, p<0.001; tactile conditioning median = 0.81, IQR=[0.66, 0.91], n=200 cells; visual conditioning median=−0.01, IQR=[−0.37, 0.43], n=120 cells). (**b**) Position of all POm cells separated by conditioning type. Cell positions are projected on the coronal plane (top row) and sagittal plane (bottom row), as in *Figure 2d and e*. Significantly responding cells are colored by their selectivity index. Gray Xs indicate unresponsive cells. Positions in the medial-lateral and anterior-posterior axes are jittered for visualization only. (**c**) Selectivity index of significantly responding cells as a function of their position in POm along the anterior-posterior (left), dorsal-ventral (middle), and medial-lateral axes (right). Light gray: tactile conditioning. Dark gray: visual conditioning. A linear model of selectivity index with experiment type and anatomical position as predictors revealed a significant effect of medial-lateral position and an interaction between position and experiment type. In visually conditioned mice, cells in the lateral portion of POm were more selective to the air puff (model p<10⁻⁶⁰, adjusted R²=0.61, conditioning p<10⁻⁵, medial-lateral p<10⁻⁴, ML-conditioning interaction p<10⁻⁴, all other terms p>0.1).

the lateral and dorsal region of POm (*Figure 6b*, right), a subregion receiving both top-down inputs from the barrel cortex and ascending input from the brainstem (*Sumser et al., 2017*; *Groh et al., 2014*). Linear regression of the selectivity index against a cell's conditioning type and anatomical position along each axis (*Figure 6c*, Methods) revealed that only the medial-lateral position was significantly correlated with the selectivity index. Thus, there appears to be a core location in the lateralmost portion of POm that is always whisker sensitive, while conditioning dictates the selectivity of the rest of POm.

## Conditioning also reshapes LP activity

Conceivably, engaging in a visual or tactile task might differentially set the baseline activity of LP and POm accordingly. We first compared spontaneous firing rates, defined as the mean rate over periods that contained no licks, stimuli, or rewards for at least 2 s and that lasted at least 6 s. Conditioning type did not statistically change mean firing rates in POm or LP, though POm cells had a higher baseline firing rate on average (*Figure 7a*).

Like POm, LP exhibited a heterogeneous set of stimulus responses that were highly dependent on conditioning (*Figure 7b–d*). For each LP cell, we determined if it responded to the air puff, grating, or both by comparing its firing rate within 1 s of stimulus onset to baseline (same analysis performed on POm cells in *Figure 3d and f*). In tactilely conditioned mice, the majority of LP cells responded to both the air puff and the drifting grating (25 cells, 39%) or the air puff alone (23 cells, 36%), with only 4 cells (6.3%) responding solely to the drifting grating. At the population level, their firing rates during the air puff stimulus period were significantly higher than the drifting grating stimulus period (*Figure 7e*, left). In visually conditioned mice (*Figure 7d*) we observed far fewer cells responding only to the air puff (6 cells, 9%), but a similar proportion of cells that responded to the drifting grating or to both stimuli (six grating responding cells, 11.9%; 30 cells responding to both stimuli, 44.7%). Similar to what we observed in POm, the magnitudes of air puff and grating responses were not significantly different (*Figure 7e*, right; *Figure 7g*). Indeed, the magnitude of the LP air puff response was significantly larger in tactilely conditioned mice than in visually conditioned mice (*Figure 7g*). Thus, tactile conditioning potentiates tactile responses in LP.

We also examined how conditioning affected low-latency stimulus responses in LP (*Figure 7—figure supplement 1*). At the population level, we observed a peak in LP population activity approximately 80 ms after visual stimulus onset (*Figure 7—figure supplement 1a*). Interestingly, this early, putatively sensory-evoked activity was apparent only in tactilely conditioned LP. LP also exhibited low-latency responses to the air puff onset in both conditioning groups, similar to air puff-evoked activity in POm. Further mirroring POm, conditioning altered LP stimulus response latency. Visually conditioned LP had far more cells with a drifting grating response latency between 100 and 500 ms compared to tactilely conditioned LP (*Figure 7—figure supplement 1b*). As a result, the mean drifting grating response latency was significantly higher in visually conditioned LP cells compared to tactilely conditioned cells (*Figure 7—figure supplement 1c*; p=0.001, Wilcoxon rank-sum test). There was no effect of conditioning on response latencies to the air puff.

As LP correlates with movement and arousal to the same degree POm does (*Petty et al., 2021*), we again computed a movement-corrected firing rate by regressing out pupil radius, whisking amplitude, and licking (*Figure 7f*). This regression quashed the difference in activity between conditioning types in the offset period only. Interestingly, we still observed an air puff response in visually conditioned LP even after this regression (*Figure 7f*, right), consistent with a strong salience of facial stimuli. We did not observe a relationship between anatomical location and visual-tactile selectivity in LP (*Figure 7—figure supplement 2*); however, our recordings were concentrated on the anterior-medial portion of LP and thus may not be characteristic of the entire nucleus. These results demonstrate that, like POm, LP activity is reshaped according to the modality of the attended stimulus.

## Discussion

Here, we investigated whether the secondary sensory thalamic nuclei differentially encode relevant and distracting stimuli. Our novel head-fixed conditioning paradigm demonstrates that mice can attend to a visual or tactile stimulus while ignoring a stimulus of the other sensory modality. The second modality neither enhanced nor suppressed action. Learning dramatically altered activity in POm: after visual conditioning, a large portion of POm cells responded to the onset of the visual stimulus. POm correlated with arousal and movement regardless of conditioning type, but movement correlations could not explain POm stimulus responses. Learning also altered LP activity: LP responses to tactile stimuli were greater after tactile conditioning, while visual responses were weaker. The predominant response in both POm and LP was to the conditioned stimulus. Thus, the secondary sensory thalamic nuclei encode the behavioral relevance of a stimulus, regardless of its modality. Reward predicting stimuli activated secondary nuclei concurrently rather than selectively according to the attended modality.

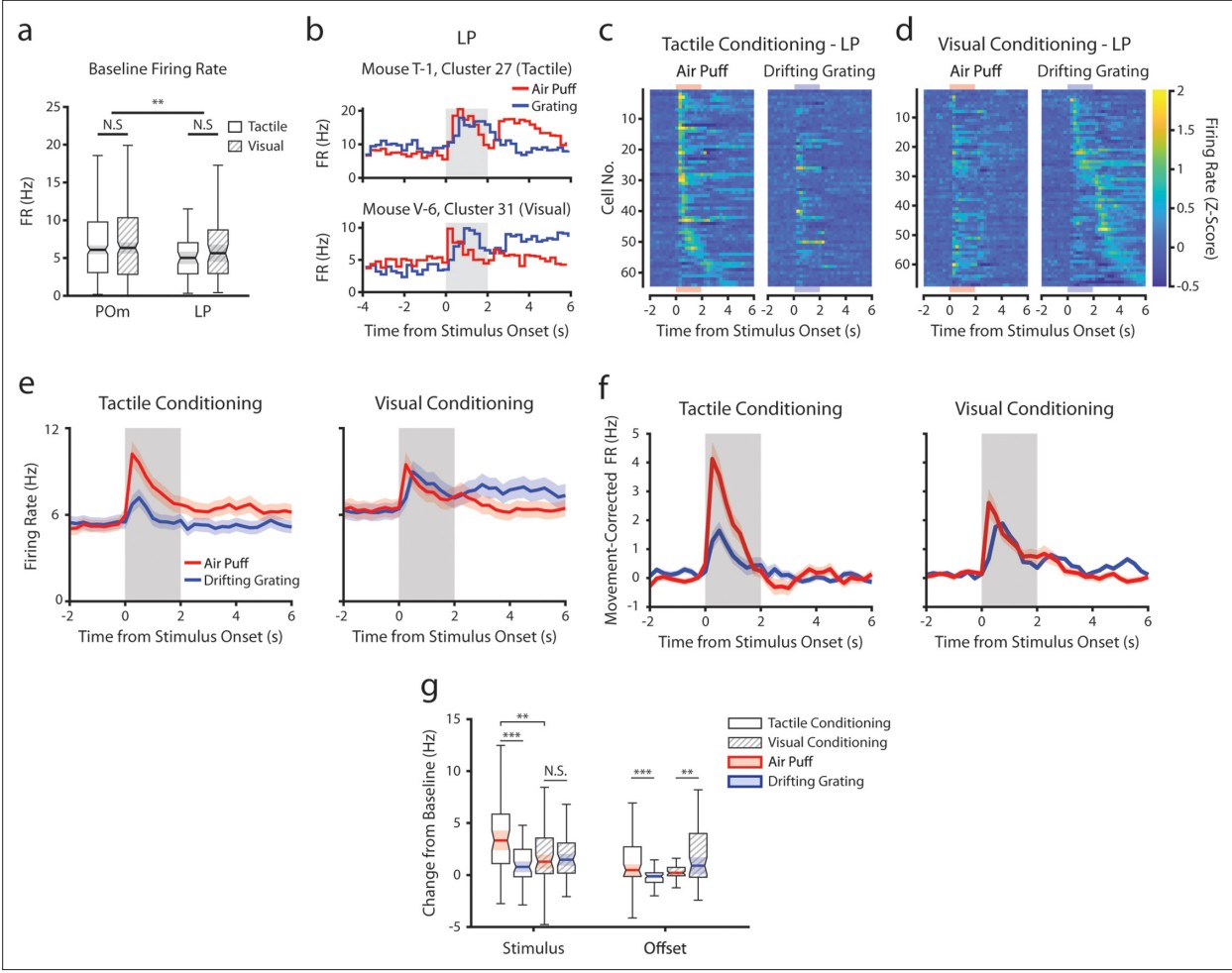

**Figure 7.** Conditioning also reshapes lateral posterior nucleus (LP) responses. (**a**) Baseline firing rates of all posterior medial nucleus (POm) and LP cells separated by conditioning type. POm cells had higher firing rates than LP cells, and firing rates did not significantly differ by conditioning type (two-way ANOVA, region $F$=8.0, p=0.0049; conditioning type $F$=0.44, p=0.51; interaction $F$=1.44, p=0.23). (**b**) Example LP cells from a tactilely conditioned mouse (top) and a visually conditioned mouse (bottom). Mean firing rates over each trial are aligned to the drifting grating (blue line) and the air puff (red line). Gray region indicates the timing of both stimuli. (**c**) Firing rate of all LP cells from tactilely conditioned mice aligned to either the air puff (left) or the drifting grating (right). Cells are sorted by the timing of the peak firing rate when aligned to the air puff. n=64 cells, 7 mice, range = 4–16 cells/ mouse, median = 7 cells/mouse. (**d**) Firing rate of all LP cells from visually conditioned mice aligned to either the air puff (left) or the drifting grating (right). Cells are sorted by the timing of the peak firing rate when aligned to the drifting grating. n=67 cells, 8 mice, range=1–19 cells/mouse, median = 6 cells/mouse. (**e**) Event-triggered average firing rates of all LP cells (mean ± SEM) aligned to either the air puff (red) or the drifting grating (blue) in tactilely conditioned mice (left) and visually conditioned mice (right). Gray region indicates the timing of both stimuli. (**f**) Movement-corrected firing rates. The firing rate of each cell was fit to a linear model with whisking, licking, pupil radius, and baseline firing rate as predictors. Model residuals of all cells (mean ± SEM) are aligned to the drifting grating and the air puff, as in panel e. (**g**) Box plots of the change in firing rate from baseline for each LP cell during the first second of stimulus onset ('stimulus' period) and during the first 2 s post-stimulus ('offset' period). In tactilely conditioned mice, the change from baseline was greater in the air puff stimulus period than the visual stimulus period (two-way ANOVA with post-hoc signed-rank test; conditioning type $F$=1.268, p=0.13; stimulus type $F$=15.7, p≤0.001; interaction $F$=13.25, p=0.003, signed-rank p<10⁻⁴). There was no difference in the change from baseline between stimulus types in visually conditioned mice (p=0.91, signed-rank test). The change from baseline in during the air puff stimulus period was also significantly greater in tactilely conditioned LP cells than in visually conditioned cells (p=0.002, rank-sum test). In both conditioning types, firing rates in the offset period were significantly different between stimulus types (two-way ANOVA, conditioning $F$=2.1, p=0.15; stimulus $F$=0.15, p=0.69; interaction $F$=19.5, p<10⁻⁴; signed-rank p<10⁻⁴ in tactile conditioning, p=0.007 in visual conditioning).

The online version of this article includes the following figure supplement(s) for figure 7:

**Figure supplement 1.** Conditioning alters stimulus response latencies in lateral posterior nucleus (LP).

**Figure supplement 2.** Lateral posterior nucleus (LP) responses to tactile and visual stimuli as a function of anatomical location.

## Learning is shaped by interactions of stimulus salience and behavioral relevance

When trained on a traditional, trial-based structure, visually conditioned mice learned that the drifting grating predicted reward and that the air puff was negatively correlated with reward. Both stimuli modified action. In our novel design, however, the unrewarded stimulus was decoupled from the reward entirely, making it a true distractor. When trained on this version of the task, visually conditioned mice learned to completely ignore the air puff. By contrast, tactilely conditioned mice always ignored the unrewarded drifting grating regardless of task structure. These mice also learned the stimulus-reward association after fewer sessions. The air puff was thus innately more salient than the drifting grating. This difference in saliency is in spite of our attempts to minimize the strength of the air puff and to match the drifting grating in terms of contrast, size, spatial frequency, and temporal frequency to typically observed mouse visual cortex receptive fields. It is thought that mice rely more strongly on their whiskers than vision for many behaviors, and that all but the subtlest of vibrissa stimuli are salient (*Warren et al., 2021*). However, the difference in saliency is more likely related to the engagement of personal space: even humans find facial tactile stimuli inherently more salient than arbitrary distant visual stimuli. Regardless of the mechanism of saliency, mice can be conditioned to completely ignore a salient stimulus in favor of a more behaviorally relevant one.

## POm responds to behaviorally relevant stimuli regardless of modality

POm is only weakly activated by single whisker deflections, typically requiring large deflections of multiple whiskers to elicit a response (*Diamond et al., 1992*; *Moore et al., 2015*; *Ahissar et al., 2000*.) These sensory responses likely arise from ascending input from the somatosensory brainstem and/or descending input from the primary somatosensory cortex (*Chiaia et al., 1991*; *Deschênes et al., 1998*; *Veinante et al., 2000*). Consistent with those studies, we observed POm cells that responded to the onset of the air puff in both conditioning groups. Tactilely conditioned mice additionally exhibited POm responses to not only to the onset of the air puff but also its entire seconds-long duration. Sustained POm responses to air puff were weaker and less common in visually conditioned mice.

The early response latencies we report here ($\leq 50$ ms) are similar to, but slightly greater than, those previously reported in anesthetized rats (15–20 ms) (*Diamond et al., 1992*). This discrepancy could be due to the difference in species, specific stimulus, or state of consciousness between these two studies. We note that, in the absence of stimulus, POm cells are almost entirely silent under anesthesia but have a high spontaneous firing rate in awake animals. Thus, it can be difficult to directly compare sensory response latencies between the two.

Remarkably, visual conditioning sensitized POm to visual stimuli. These visual responses largely resembled the air puff responses in tactilely conditioned mice, with individual cells' activities tiling the entire duration of the stimulus. Visual responses in POm are unexpected. Though it receives excitatory input from a wide array of cortical regions (*Liao et al., 2010*; *Aldes, 1988*; *Gharaei et al., 2020*), POm is not known to receive direct input from the retina or visual cortex. Rather than a sensory response per se, visual-evoked activity likely reflects encoding of behavioral state or attention. Consistent with that idea, many visually responsive cells also responded to the air puff. It could be that POm cells respond to any increase in attention, be it due to the unconditioned but salient, 'attention-grabbing' air puff or the subtler but now behaviorally relevant drifting grating.

There is growing evidence that POm is a heterogeneous region composed of multiple subnuclei defined by distinct thalamocortical outputs and corticothalamic inputs (*El-Boustani et al., 2020*; *Sumser et al., 2017*; *Groh et al., 2014*; *Mease et al., 2016*; *Viaene et al., 2011*). We found that the effects of conditioning on POm stimulus responses varied by anatomical location. While visual conditioning induced visual responses throughout POm, cells in the lateral dorsal region of POm remained more sensitive to the air puff than the drifting grating. Layer 5a cells in the S1 barrel field preferentially project to this region of POm, and POm cells receiving S1 corticothalamic input also receive convergent projections from the whisker-sensitive brainstem nucleus SpVi (*Sumser et al., 2017*). It is likely that the observed activity in lateral dorsal POm is driven by true whisker responses in SpVi and S1.

## Movement does not explain POm plasticity

A recent study demonstrated that, in a tactile Go-NoGo paradigm, POm activity correlates with a mouse's decision to lick or withhold licking (*El-Boustani et al., 2020*) and similar results have been

demonstrated in a detection task using the forepaw (*La Terra et al., 2022*). Decision, response, and reward consumption coincide with movement – including whisking and licking. Responses in POm might instead be movement signals, though we have previously demonstrated that POm-whisking correlations cannot be fully explained by sensory reafference nor motor efference copy from S1, M1, or the superior colliculus (*Petty et al., 2021*). We found here that POm activity correlated with whisking amplitude, pupil diameter, and licking regardless of conditioning.

In both conditioning groups, POm activity was elevated during reward presentation and consumption, coincident with licking. POm receives somatotopic projections from all of S1, not just the barrel cortex (*Diamond et al., 1992*; *Fabri and Burton, 1991*). As such, stimulation of the mouth and tongue - as occurs when the mouse licks - could contribute (via S1) to late phases of POm activity. Nevertheless, tactilely conditioned mice had a greater correlation between licking and POm activity. This suggests an increased sensitivity to all tactile stimulation as mice attend to the air puff, compared to mice that ignore the air puff and attend to the drifting grating.

Furthermore, visually conditioned mice had a greater correlation between POm and pupil radius, but only at long time lags (>500 ms). Conceivably, as mice attend to visual stimuli, they could be more sensitive to all changes in retinal input. However, at time lags close to 0 ms there is no difference in POm-pupil correlation between tactile and visual conditioning groups, and we observe very few visually responsive cells in tactilely conditioned mice. Therefore, this activity may instead represent an increased correlation between POm and arousal. The drifting grating is inherently less salient than the air puff; consequently, the visual conditioning task may require greater attentional demands, which in turn could further engage the secondary thalamic nuclei.

As licking, whisking, and pupil dilation are all significantly correlated with POm, we examined whether this relationship could explain the stimulus-evoked activity. Even after we factored out the possible contributions of these movement variables using regression, POm responses to the onset of conditioned stimuli were stronger than those of the unconditioned stimuli. In both tactilely and visually conditioned mice, movement could not explain the increased firing rate at air puff onset. These low-latency responses across conditioning groups are likely due in part to 'true' sensory responses driven by S1 and SpVi. In tactilely conditioned mice, movement regression attenuated, but did not eliminate, activity in the later portion of the air puff. Activity in the rewarded offset period, however, was suppressed almost entirely, suggesting that post-stimulus activity is due to non-sensory signals when mice lick the water port.

## LP activity is similarly shaped by conditioning

Like POm, LP displayed varied stimulus-evoked activity that was heavily dependent on conditioning. LP responded to the air puff robustly and with low latency, despite lacking direct somatosensory inputs. Though LP activity correlates with whisking and air puffs induce whisking, the movement could not explain such responses in either conditioning group. The air puff response was larger in tactilely conditioned mice, indicating that, like POm, LP is more responsive to the occurrence of any behaviorally relevant stimuli regardless of modality.

Unlike POm, LP responded to the drifting grating in both conditioning groups. LP receives input from the retina, the superior colliculus, and several visual cortical areas (*Roth et al., 2016*; *Bennett et al., 2019*; *Leow et al., 2022*; *Born et al., 2021*; *Juavinett et al., 2020*; *Scholl et al., 2021*). In tactilely conditioned mice, 45% of LP cells responded to the onset of the drifting grating, consistent with previous studies of LP responses to simple visual stimuli (*Allen et al., 2016*). In visual conditioning animals, a greater portion of cells were visually responsive (56%), suggesting the existence of a population of LP cells that are not sensitive to simple visual stimuli but are active when a mouse is aroused and attentive. This is further reflected in the increased latency of drifting grating responses after visual conditioning.

It is notable that a majority of LP cells had significant responses to the air puff in both conditioning groups. This is despite the fact that LP has no known inputs from the somatosensory brainstem or the barrel cortex. However, we have previously seen that LP activity correlates with whisking just as strongly as POm does, and at a time lags less than 50 ms (*Petty et al., 2021*), a fact which we attribute to LP tracking global arousal. Furthermore, here we observed that the air puff is inherently more salient to naive mice than the drifting grating. It is possible that even after visual conditioning, the air puff remains salient and arousing but the mice have learned not to respond to it. These responses

could be the result of neuromodulators, such as norepinephrine or acetylcholine, acting either directly on LP or on its inputs.

LP has several subregions with distinct cytoarchitecture, anatomical inputs, and retinotopic maps (*Nakamura et al., 2015*; *Bennett et al., 2019*; *Leow et al., 2022*; *Takahashi, 1985*; *Zhou et al., 2017*). Our recordings were clustered in the anterior and medial portion of LP, which receives input primarily from the frontal cortex and only sparse input from V1 and the superior colliculus. Other regions of LP may receive greater input from the visual cortex, and thus be more responsive to visual stimuli and less affected by conditioning, similar to the dorsal-ventral region of POm we identified. Further characterization of the various LP subregions in behavioral contexts could reveal how specific cortical and subcortical inputs contribute to attention.

## Possible mechanisms of non-sensory responses

What might be driving these non-sensory responses in the secondary nuclei? One likely mechanism is neuromodulation. Cortical levels of acetylcholine and norepinephrine track arousal (*Eggermann et al., 2014*; *Reimer et al., 2016*). Both acetylcholine and norepinephrine can act directly on thalamic nuclei like VPM (*Aguilar and Castro-Alamancos, 2005*; *Hirata and Castro-Alamancos, 2010*; *Varela, 2014*) and may do so on POm and LP as well. Acetylcholine can also enhance POm tactile responses by suppressing GABAergic activity in the zona incerta (*Masri et al., 2006*). LP receives inhibitory input from the zona incerta (*Barthó et al., 2002*), so LP visual responses are likely modulated by cholinergic activity in the same way. Either direct modulation or indirect disinhibition could underlie enhanced air puff responses in tactilely conditioned POm and enhanced drifting grating responses in visually conditioned LP. State-based disinhibition of POm could also contribute to 'visual' responses after visual conditioning by potentiating non-sensory inputs to POm.

Another potential culprit is direct corticothalamic input. S1 is thought to be the primary driver of POm (*Diamond et al., 1992*). Though S1 activity alone does not drive arousal-related activity in POm, direct S1-to-POm projections contribute to the performance of tactile discrimination tasks (*Qi et al., 2022*). Glutamatergic S1 inputs could underlie enhanced air puff responses in POm in behavioral contexts; however, S1 activity cannot explain the same effect in LP. Instead, visual responses in POm and tactile responses in LP might be driven by high-order cortex. In nonhuman primates, the lateral pulvinar has been shown to facilitate communication between the visual region V4 and higher-order cortical regions. These include the lateral intraparietal area (*Eradath et al., 2021*) and the temporo-occipital area (*Saalmann et al., 2012*), regions that are engaged in visual attention tasks. Indeed, the secondary nuclei have been considered possible hubs for cortico-cortical communication (*Zhou et al., 2016*; *Lohse et al., 2021*; *Theyel et al., 2010*) and are known to be innervated by the posterior parietal cortex. Elevated activity in POm and LP could reflect increased cortico-cortical communication between high-order regions, primary sensory regions, and motor regions based on attentional needs.

## The role of high-order thalamus in attention

The secondary visual thalamic nucleus in primates – the lateral pulvinar – has been known to play a role in guided visual attention. In humans, pulvinar damage causes a variety of attentional deficits, impairing the ability to selectively attend to a visual stimulus in the presence of a distractor (*Snow et al., 2009*; *Ward et al., 2002*). In nonhuman primates, chemical silencing of the pulvinar induces spatial neglect (*Wilke et al., 2010*; *Zhou et al., 2016*). Furthermore, pulvinar responses to a visual stimulus are strongest when a monkey directs attention toward that stimulus (*Petersen et al., 1985*). Similar experiments have not yet been performed for rodent LP, nor have they been directly extended to the somatosensory thalamus.

A consistent result of attention studies in humans and nonhuman primates is the enhancement of cortical and thalamic sensory responses to an attended visual stimuli (*Zhou et al., 2016*; *Gregoriou et al., 2009*; *Moran and Desimone, 1985*). Here, we show not just enhancement of responses to sensory stimuli within a single modality, but also across modalities. It is worth investigating further how the secondary thalamus and high-order sensory cortex encode attention to stimuli outside of their respective modalities. Our surprising conclusion that the nuclei are equivalently activated by behaviorally relevant stimuli is nevertheless compatible with these previous studies.

## The role of secondary thalamus in learning and cortical plasticity

Experience-dependent synaptic plasticity is a substrate of learning and memory. In anesthetized mice, POm excitation can facilitate long-term plasticity (LTP) in layer 2/3 pyramidal cells in S1, while POm silencing blocks those same plasticity effects (*Gambino et al., 2014*). Similarly, whisker-dependent associative learning potentiates POm-to-S1 synapses (*Audette et al., 2019*). In the visual cortex, excitation of layer 2/3 neurons paired with stimulation of layer 4 cells induces LTP in that layer 2/3 cells (*Kirkwood and Bear, 1994*). LP thus likely contributes to plasticity in the visual cortex. Elevated firing rates in POm and LP could contribute to learning the behavioral relevance of novel stimuli. That is, they may open a window of plasticity during periods of high arousal and task engagement. As such, we expect that disruption of thalamocortical output from secondary nuclei to the sensory cortex will disrupt or even prevent associative learning. Based on our findings, disruption of POm activity could impair even visual learning, and disruption of LP activity tactile learning, particularly if there were a strong multimodal component in the task. Such future experiments would further illuminate how the secondary nuclei participate in sensory processing and could pave the way for further studies of thalamocortical and corticothalamocortical circuits.

## Conclusion

Our study demonstrates that a secondary sensory thalamic nucleus is dramatically modulated by behaviorally relevant events both in its modality as well as other modalities. Thus, reward-predicting stimuli modulate secondary nuclei concurrently, rather than selectively, according to the attended modality. This broad dependence on task engagement across secondary nuclei suggests a potential role in gating cortical plasticity and overall attention, particularly with regard to multimodal associative learning and transthalamic communication between cortical areas.

## Methods

**Key resources table**

| Reagent type (species) or resource | Designation | Source or reference | Identifiers | Additional information |
|---|---|---|---|---|
| Software, algorithms | MATLAB | MathWorks, Inc. | v2022b | |
| Software, algorithms | Kilosort | https://github.com/MouseLand/Kilosort (*Pennington et al., 2024*) | v3 | |
| Software, algorithms | Phy | https://github.com/cortex-lab/phy | v2.01b | |
| Software, algorithms | Arduino IDE | https://www.arduino.cc/ | v1.8.x | |
| Software, algorithms | PsychoPy | https://github.com/psychopy/psychopy/releases/tag/2020.1.1 (*Peirce, 2020*) | v2020.1.1 | |
| Software, algorithms | FIJI | https://imagej.net/software/fiji | N/A | |
| Software, algorithms | SHARP-Track | https://github.com/cortex-lab/allenCCF | N/A | |
| Chemical compound, drug | DiI (1,1'-Dioctadecyl-3,3,3',3'-tetramethylindocarbocyanine perchlorate) | Sigma-Aldrich | CAS# 41085-99-8 | |
| Chemical compound, drug | Streptavidin, Alexa 647 Conjugate | ThermoFisher | Cat# S21374 | |

### Animals

Twenty-three wild-type C57Bl/6 (4 female, 19 male) were used in this study. Mice were either purchased from Jackson Laboratories or bred in our own colony, and were between 12 and 24 wk old. All experiments were conducted under the supervision and approval of the Columbia University Institutional Animal Care and Use Committee (protocol number AC-AABP0555).

### Surgery

Mice were administered the global analgesics carprofen (5 mg/kg) and buprenorphine (0.05–0.1 mg/kg) subcutaneously, and anesthetized with isoflurane. The scalp was injected with the local analgesic

bupivacaine (5 mg/kg). Mice were then placed in a stereotax, and the scalp and fascia were removed. A custom-designed stainless steel head plate was affixed to the skull with Metabond. Carprofen was administered subcutaneously 24 hr post-surgery. Mice were allowed to recover for 7 d, during which additional carprofen was administered as deemed necessary. After recovery, mice were deprived of water, habituated to the behavioral apparatus, and conditioned (*Behavior Apparatus* and *Conditioning*, below). After behavioral training, mice were again treated with carprofen and buprenorphine, anesthetized with isoflurane, and placed in a stereotax. A ~200 μm-wide opening was cut on the left side of the skull, centered at 1.8 mm posterior to bregma and 1.5 mm lateral of the midline. This opening was sealed with Kwik-Cast. A ground pin was inserted over the frontal cortex of the right hemisphere. Mice were allowed to recover for 24 hr, after which recordings took place. At the end of the experiments, mice were deeply anesthetized with isoflurane, then perfused transcardially with phosphate buffer followed by 4% paraformaldehyde.

## Behavior apparatus

The behavior apparatus was contained in a black box with a light-blocking door. It was constructed from metal posts on an aluminum breadboard (Thorlabs). Mice were held in a 3D-printed hutch and head-fixed with a machined steel head plate holder. Two-second-long air puffs (0.5–1 bar) were delivered through a nozzle (cut p1000 pipet tip, approximately 3.5 mm diameter aperture) and controlled by a solenoid. Water rewards (8–16 μL) were similarly delivered by a solenoid. Licking was recorded with an infrared proximity detector. The visual stimulus was presented on a monitor (5 inch, 800×480 resolution, Eyoyo) positioned 11 cm from the right side of the mouse hutch occupying 52°x 33° of visual space. The stimulus consisted of vertical drifting bars (spatial frequency 0.05°/cycle, temporal frequency 1.5 Hz) with maximum contrast (maximum luminance 247 cd/m$^2$, minimum luminance 3.2 cd/m$^2$) and lasted 2 s. When the stimulus was off the monitor displayed a gray background set to the same mean luminance as the stimulus (125 cd/m$^2$). A photodiode was used to detect the onset of the visual stimulus to synchronize it with the rest of the apparatus.

An Arduino UNO microcontroller drove the two solenoids. A digital signal from the Arduino was used to trigger the visual stimulus, which was driven with a custom Python script (PsychoPy). The solenoid TTL signals, the lick detector analog signal, and the photodiode output were all recorded with OpenEphys hardware and software (see 'Electrophysiology' below). When capturing video, the Arduino also drove a blinking infrared LED to synchronize video data.

## Conditioning

Throughout habituation and conditioning, mice were deprived of water in their home cages and received all daily water intake while placed in the behavior apparatus. We closely monitored their weight and health, and ad libitum water was provided if their weight decreased below 80% of baseline.

Prior to conditioning, mice were habituated to handling and head fixation. They were placed in the behavior apparatus over several sessions of increasing duration, spanning 10–30 min, during which they were given their water ration. Mice were habituated for at least three such sessions over the course of 2 to 3 d. During habituation, mice were observed and evaluated for signs of discomfort or distress (i.e. squeaking and trying to escape). Mice were considered habituated if they completed a 30 min session without displaying any such behaviors, and all mice reached habituation after 3 d.

Conditioning began 24 hr after the mice reached habituation. In the first phase of conditioning ('shaping'), mice were separated into two cohorts: a 'tactile' cohort and a 'visual' cohort. Mice were presented with tactile stimuli (a 2 s-long air puff delivered to the right whisker field) and visual stimuli (a 2-s playback of a vertical drifting grating on a monitor, positioned on the right side of the face). Throughout conditioning, mice were monitored via webcam to ensure that the air puff only contacted the distal tips of the whiskers and did not disturb the facial fur nor cause the mouse to blink, flinch, or otherwise react – ensuring the stimulus was not aversive. The stimulus types were randomly ordered. In the visual conditioning cohort, the visual stimulus was paired with a water reward (8–16 μL) delivered at the time of stimulus offset. In the tactile conditioning cohort, the reward was instead paired with the offset of the air puff. Regardless of the type of conditioning, stimulus type was a balanced 50:50. During the shaping version of the task, stimuli were delivered with an inter-stimulus interval (ISI) of 8–12 s (uniform distribution). This distribution changed in the full version of the task (see below).

Mice were trained for one session per day, with each session consisting of an equal number of visual stimuli and air puffs. Sessions ranged from 20 to 60 min and about 40–120 of each stimulus.

Throughout training and during electrophysiology recordings, we recorded analog signals of mouth and tongue movements with an infrared proximity detector located below the water reward spout. Individual licks were detected by manually inspecting this signal and setting a threshold using custom MATLAB code. Throughout conditioning, we evaluated performance by measuring the anticipatory licking behavior of a mouse to a rewarded stimulus or unrewarded stimulus. For each stimulus presentation, we counted the number of times a mouse licked in the 2 s prior to the stimulus onset and the number of licks during the 2 s stimulus and calculated a lick index (LI) for that stimulus (*Jurjut et al., 2017*).

$$LI = \frac{licks_{stimulus} - licks_{baseline}}{licks_{stimulus} + licks_{baseline}}$$

A positive lick index indicates that a mouse responds to a stimulus by licking more and a negative index indicates that a mouse responds by withholding licking. Licks that occurred within 50 ms of the stimulus onset were excluded when computing the lick index.

Mice were trained on the shaping version of the task until the mean lick index of the CS+ for a given session was significantly greater than 0 (sign rank test), with a minimum of 3 d of conditioning. After reaching this threshold, mice were switched to the 'full' version of the task. In the full task, the stimuli and reward were identical, but stimuli were presented at uncorrelated and less predictable intervals. For a given stimulus, the time until the next stimulus of the same type was drawn randomly from an exponential distribution with a mean of 10 s plus an offset of 8–12 s (drawn from a uniform distribution), for a mean ISI of 20 s. The maximum ISI was capped at 55 s, and ISIs were binned by seconds. Both the air puff and the visual stimulus ISIs were drawn from the same distribution but were done so independently. Thus, the stimuli could overlap, the timing of the unconditioned stimulus was uncorrelated with the CS+ or the reward, and each stimulus had a flat hazard rate for the majority of possible ISIs. Mice were then trained on the full version of the task until the mean LI per session for the unconditioned stimulus was not significantly different from 0 (signed-rank test) for a minimum of 4 d.

## Electrophysiology

We performed simultaneous recordings from LP and POm from fully trained mice as the animals performed the full version of the task. Using a motorized micromanipulator (Scientifica PatchStar), we inserted a 64-channel electrode array (Cambridge Neurotech, models H3 and H9) into the craniotomy. Arrays were inserted vertically to a depth of 3.1–3.5 mm from the cortical surface (mean 3.4 mm). Prior to recording, the tip of the array was dipped into a DiI solution to label recording sites. Array recordings were acquired using an OpenEphys amplifier with two digital headstages (Intan) at 30 kHz and with a bandwidth of 2.5 Hz to 7.6 kHz. The same acquisition system was used to record stimulus timing, licking, and a syncing signal for the video.

We used KiloSort3 to detect spikes and assign them to putative single units (*Pachitariu et al., 2016*) and Phy2 (https://github.com/cortex-lab/phy; *Rossant, 2024*; *Rossant et al., 2016*) to manually inspect each unit. We merged units that appeared to originate from the same cell, based on waveform shape, auto- and cross-correlations, and firing rate over time. We excluded cells that lacked a refractory period (i.e. a dip in autocorrelation within 3 ms). Specifically, units with more than 10% of spikes having inter-spike intervals shorter than 3 ms were considered multi-unit and excluded. Units were assigned to the array channel on which its mean waveform amplitude was largest.

## Histology

We sectioned paraformaldehyde-fixed brains into 100 μm sections using a vibratome. To visualize the border between POm and VPM, we stained sections for endogenous biotin using fluorescently conjugated streptavidin (Alexa Fluor 647, Thermo Fisher). We imaged the sections on a slide scanner (Nikon AZ100 Multizoom). We then used SHARP-Track to align images to the Allen reference atlas and to identify the position of the electrode array by tracing the DiI track (*Shamash et al., 2018*, https://github.com/cortex-lab/allenCCF; *Shamash et al., 2024*). After histological alignment (see below) we used the depth of each putative cell along the array to assign it to a brain region. When assigning cells to brain regions, we excluded cells that were within 50 μm of a region boundary.

## Videography

During electrophysiology recordings, two PS3 Eye cameras were used to capture video of the mouse's eye and whiskers. The whisker camera was positioned below the mouse's head and recorded at 125 fps. The eye camera was positioned on the right side of the face and recorded at 60 fps. The OpenEphys Bonsai program was used to record video.

Pupil size was measured from video offline using custom MATLAB code. We defined a region of interest around the eye and applied a threshold to each video frame to create a binary image of the pupil. We then used the function 'region props' to fit an ellipse to the pupil and estimated pupil radius as the geometric mean of the semi-major and semi-minor axes of that ellipse. The trace of pupil radius over time was smoothed over five frames (83 ms). Large discontinuities in the pupil radius vector (such as those caused by a blink) were excluded based on a median filter (spanning 3 s) and linearly interpolated.

Whiskers were automatically tracked from videos using the software package *Trace* (*Clack et al., 2012*). Custom MATLAB software was used to compute the mean angle of all whiskers over each frame. The mean angle was bandpass filtered from 4 to 30 Hz and passed through a Hilbert transform to calculate phase. We defined the upper and lower envelopes of the unfiltered median whisking angle as the points in the whisk cycle where phase equaled 0 (most protracted) or ±π (most retracted), respectively. Whisking amplitude was defined as the difference between these two envelopes.

A blinking LED paired with a TTL pulse was used to synchronize the video frames to the data collected via the OpenEphys acquisition system. After processing the videos, this signal was used to align relevant video data (e.g. pupil radius and whisking amplitude) to physiology and behavioral data. Video vectors were linearly interpolated from their respective frame rates to 1 kHz. Stimulus and licking signals were downsampled to 1 kHz, and spike times were rounded to the nearest millisecond.

## Stimulus responses

We examined each cell's response to the two task stimuli by comparing its firing rate during a 'stimulus period' to its baseline firing rate. We first excluded periods in the recording with overlapping stimuli, defined as any stimulus occurring within 6 s of a stimulus of a different type. We then counted the number of spikes that occurred within 1 s prior to the onset of each stimulus (baseline period) and within 1 s of the stimulus onset (stimulus period). We classified a cell's response as significant by first performing an ANOVA between the baseline firing rates, the rates during the air puff stimulus period, and the rates during the drifting grating stimulus period over each stimulus presentation. This was followed by a Holm-Bonferroni multiple comparisons correction, which was applied to cells within each experimental group (i.e. p values from tactile conditioning POm cells were corrected separately from p values of visual conditioning POm cells, and likewise for LP cells). For each cell with a corrected ANOVA p<0.05, we performed a paired Wilcoxon rank-sum test between the air puff stimulus firing rates and the corresponding baseline firing rates. Cells with a Wilcoxon p<0.05 were classified as 'air puff-responding.' We likewise classified cells as 'grating responding' with a Wilcoxon test between the grating stimulus period firing rates and the corresponding baseline firing rates. No correction was applied to the ranked-sum values. Any cell that responded to either or both stimulus is deemed 'significantly responding'.

For significantly responding cells, we computed a stimulus selectivity index (SI). The selectivity index measures the difference between the magnitudes of the air puff response and the drifting grating responses compared to the baseline firing rate:

$$SI = \frac{|FR_A - FR_B| - |FR_G - FR_B|}{|FR_A - FR_B| + |FR_G - FR_B|}$$

where $FR_B$ is the mean baseline firing rate, $FR_A$ is the mean firing rate during the air puff stimulus period (1 s period starting from stimulus onset) and $FR_G$ is the mean firing rate during the drifting grating stimulus period. A positive selectivity index indicates that the cell has a larger response to an air puff, in terms of absolute change in firing rate, than it does to the visual stimulus, and vice-versa for a negative index.

To determine if the anatomical location had a significant effect on the selectivity index, we fit a linear model between the selectivity index of all POm cells and each cell's position in the dorsal-ventral (DV), medial-lateral (ML), and anterior-posterior (AP) position, along with the conditioning

type (CT) of that cell. We also considered interactions between experiment type and position, but not between the different positional dimensions. The model thus had the following design:

$$SI \sim 1, \, DV, \, ML, \, AP, \, CT, \, CT*DV, \, CT*ML, \, CT*AP$$

To measure sensory response latency, we calculated the firing rate of each cell over 10 ms bins. We measured the mean firing rate of each cell aligned to either the air puff or the visual stimulus. We then computed the mean and standard deviation of the cell's firing rate 1 s prior to the stimulus onset (baseline firing rate). We defined a sensory response as the first instance where a cell's activity increased or decreased to cross a 99% confidence interval (2.576 standard deviations) of the baseline firing rate for at least two consecutive time bins, or the first time a cell's activity crossed a 99.99% confidence interval (3.891 standard deviations) for a single time bin, whichever occurred first. Baseline firing rates were defined as the average rate over 2 s prior to stimulus onset.

## Regression of movement and arousal variables

When fitting linear models, we first binned firing rates, lick rates, whisking, and pupil radius into 250 ms time bins. This bin size is long enough to capture a range of firing rates (i.e. most cells had multiple spikes per bin) while preserving the timing of licking and whisking. We tested models with bin sizes of 50, 100, and 500 ms and found qualitatively similar results. For each cell, we fit a linear model of firing rate to whisking amplitude, pupil radius, and lick rate using least-squares regression. We extracted the model residuals for each cell, which correspond to a baseline-subtracted and movement-corrected firing rate.

## Acknowledgements

The authors thank Dahee Jang, Katy Willard, Candice Lee, Ziyue Liu, and Armin Lak for comments on the manuscript; Chris Rodgers and Sam Benezra for advice on creating the behavioral apparatus; and David Park for help training mice in pilot experiments. Support was provided by a Wellcome Trust Discovery Award (225210/Z/22/Z), an Academy of Medical Sciences Professorship, NIH/NINDS R01 NS069679 and R01 NS094659, and NIH/NEI T32 EY013933.

## Additional information

### Funding

| Funder | Grant reference number | Author |
| --- | --- | --- |
| Academy of Medical Sciences | APR6\1007 | Randy M Bruno |
| Wellcome Trust | 225210/Z/22/Z | Randy M Bruno |
| National Institute of Neurological Disorders and Stroke | R01 NS069679 | Randy M Bruno |
| National Institute of Neurological Disorders and Stroke | R01 NS094659 | Randy M Bruno |
| National Eye Institute | T32 EY013933 | Gordon H Petty |

The funders had no role in study design, data collection and interpretation, or the decision to submit the work for publication. For the purpose of Open Access, the authors have applied a CC BY public copyright license to any Author Accepted Manuscript version arising from this submission.

### Author contributions

Gordon H Petty, Conceptualization, Data curation, Software, Formal analysis, Investigation, Visualization, Methodology, Writing – original draft, Writing – review and editing; Randy M Bruno,

Conceptualization, Formal analysis, Supervision, Funding acquisition, Methodology, Writing – original draft, Writing – review and editing

## Author ORCIDs
Gordon H Petty ⬡ https://orcid.org/0000-0002-2215-6714
Randy M Bruno ⬡ https://orcid.org/0000-0002-5122-4632

## Ethics
All experiments were conducted under the supervision and approval of the Columbia University Institutional Animal Care and Use Committee.(protocol number AC AABP0555).

Reviewer #1 (Public review): https://doi.org/10.7554/eLife.97188.3.sa1
Reviewer #2 (Public review): https://doi.org/10.7554/eLife.97188.3.sa2
Reviewer #3 (Public review): https://doi.org/10.7554/eLife.97188.3.sa3
Author response https://doi.org/10.7554/eLife.97188.3.sa4

## Additional files

### Supplementary files
• MDAR checklist

### Data availability
Neural and behavioral data from this study are available for download on Dryad: https://doi.org/10.5061/dryad.4j0zpc8n0.

The following dataset was generated:

| Author(s) | Year | Dataset title | Dataset URL | Database and Identifier |
|---|---|---|---|---|
| Bruno RM, Petty GH | 2024 | Data from: Attentional modulation of secondary somatosensory and visual thalamus of mice | https://doi.org/10.5061/dryad.4j0zpc8n0 | Dryad Digital Repository, 10.5061/dryad.4j0zpc8n0 |

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
