## [Editor Report · eLife Assessment]

This study provides an **important** re-evaluation of modality-specific information processing in the thalamus of trained mice. Using an elegant task design that probes competing tactile and visual stimuli, the authors present **compelling** evidence that behavioral training reshapes the sensitivity of higher-order thalamic nuclei. Despite the powerful task design and the significance of the main findings, the origin of the cross-modal responses remains an open question and requires future investigation.

---

## [Referee Report · Reviewer #1 (Public review)]

Petty and Bruno investigate how response characteristics in the higher-order thalamic nuclei POm (typically somatosensory) and LP (typically visual) change when a stimulus (whisker air puff or visual drifting grating) of one or the other modality is conditioned to a reward. Using a two-step training procedure, they developed an elegant paradigm, where the distractor stimulus is completely uninformative about the reward, which is reflected in licking behavior of trained mice. While the animals seem to take on to the tactile stimulus more readily, they can also associate reward with the visual stimulus, ignoring tactile stimuli. In trained mice, the authors recorded single unit responses in both POm and LP while presenting the same stimuli. The authors first focused on POm recordings, finding that in animals with tactile conditioning POm units specifically responded to the air puff stimulus but not the visual grating. Unexpectedly, in visually conditioned animals, POm units also responded to the visual grating, suggesting that the responses are not modality-specific but more related to behavioral relevance. These effects seem not not be homogeneously distributed across POm, whereas lateral units maintain tactile specificity and medial units respond more flexibly. The authors further ask if the unexpected cross-modal responses might result from behavioral activity signatures. By regressing behavior-coupled activity out of the responses, they show that late activity indeed can be related to whisking, licking and pupil size measures. However, cross-modal short latency responses are not clearly related to animal behavior. Finally, LP neurons also seem to change their modality-specificity dependent on conditioning, whereas tactile responses are attenuated in LP if the animal is conditioned to visual stimuli.

The authors make a compelling case that POm neurons are less modality specific than typically assumed. The training paradigm, employed methods and analyses are to the point, well supporting the conclusions. The findings importantly widen our understanding of higher-order thalamus processing features with flexibility to encode multiple modalities and behavioral relevance. The results raise many important questions on the brain-wide representation of conditioned stimuli. E.g. how specific are the responses to the conditioned stimuli? Are thalamic cross-modal neurons recruited for the specific conditioned stimulus or do their responses reflect a more global shift of attention from one modality to another? Are these cross-modal responses tracking global arousal/attention features, or actually encoding a different stimulus?

The authors clarified a number of points in the updated version of the manuscript and expanded analyses and methods descriptions, which substantially improved the paper. The different time periods around the stimuli are more clearly assigned now and make the conclusions stronger.

Especially the discussion is now well rounded and addresses the major points.

To ask if the cross-modal activity is in some way functional for task performance I would like to see if (population) activity in the classical vs. cross-modal nucleus is predictive of lick latency or frequency on a trial-to-trial basis.

I accept that the authors cannot differentiate between bottom-up "raw" sensory responses and top-down context/attention/etc signals and thus support the decision to restrict the analyses to either the likely sensory early part following stimulus onset or the (as shown here mostly movement-driven) offset period after cessation of the stimulus. However, the composite responses over different stimuli and conditioning types seem triphasic to me. I find the "ongoing" activity differences (~100-2000 ms) depending on conditioning type quite interesting and would welcome a more specific discussion on the different response periods.

Overall a very elegant and well-presented study.

---

## [Referee Report · Reviewer #2 (Public review)]

This manuscript by Petty and Bruno delves into the still poorly understood role of higher-order thalamic nuclei in the encoding of sensory information by examining the activity in the Pom and LP cells in mice performing an associative learning task. They developed an elegant paradigm in which they conditioned head-fixed mice to attend to a stimulus of one sensory modality (visual or tactile) and ignore a second stimulus of the other modality. They recorded simultaneously from POm and LP, using 64-channels electrode arrays, to reveal the context-dependency of the firing activity of cells in higher-order thalamic nuclei. They concluded that behavioral training reshapes activity in these secondary thalamic nuclei. The authors brought new analyses and figures which greatly improve their manuscript and support their conclusion. The manuscript benefits now from a better communication about both the methodology and the results. I have no more major concerns, but I feel that the readability of the manuscript could be improved with the following revisions.

Strengths

The authors developed an original and elegant paradigm in which they conditioned head-fixed mice to attend to a stimulus of one sensory modality, either visual or tactile and ignore a second stimulus of the other modality. As a tactile stimulus, they applied gentle air puffs on the distal part of the vibrissae, ensuring that the stimulus was innocuous and therefore none aversive which is crucial in their study.

It is commonly viewed that first-order thalamus performs filtering and re-encoding of the sensory flow; in contrast the computations taking place in high-order nuclei are poorly understood. They may contribute to cognitive functions. By integrating top-down control, high-order nuclei may participate in generating update models of the environment based on sensory activity; how this can take place is a key question that Petty and Bruno addressed in the present study.

Weaknesses

(1) It's difficult when reading the text to understand which results were quantified and which were not, in part because mean data as well as (s.e.m. or S.D.) do not appear either in the main text nor in the legends of the figures. Only vague and unquantified data are given in the main text. I understand that the authors may want to make the main text less heavy, but having these data fully written somewhere (i.e., main text, summary table, figure legends) rather than having to estimate through looking at a graph (especially when the data are constraint in the first 20% of the graph (Figure 4c)), would greatly improve the text's clarity and precision.

For instance, Line #173, "At the population level, POm cells in both conditioning groups had a peak of activity 40ms after air puff onset (Figure 4a)." Is this 40 ms a result of quantified data, then a s.e.m. would be informative, or a reading measurement on the Figure 4a graphs? As it stands, it is too vague a value.

(2) The authors give clearer definition of what they analyzed, which greatly improved the readability of the manuscript. The clarity of the manuscript could still be improved by solving remaining ambiguities about sensory- versus non-sensory-responses to the applied stimuli throughout the manuscript, in order to better convey the authors' conclusion that behavioral training reshapes activity in these secondary thalamic nuclei which then may participate in generating update models of the context in which the animal is performing the task.

Line #24 in the abstract "In mice trained to respond to tactile stimuli and ignore visual stimuli, POm was robustly activated by touch and largely unresponsive to visual stimuli". The abstract would better reflect the manuscript conclusions indicating that POm was robustly activated during tactile stimuli.

(3) The new analysis of the "early" responses in Pom cells pointed out, Line #173, that "At the population level, POm cells in both conditioning groups had a peak of activity 40ms after air puff onset (Figure 4a)." Previous works cited by the authors, Diamond et al. (1992), described tactile responses in Pom cells at 15-20ms latency which were suppressed by the barrel cortex inactivation.

The 40ms-latency responses described in this manuscript therefore do not fit with "purely sensory" and barely with S1-feedbacks, as proposed on line #168 "Such responses could be "purely sensory" (i.e. driven by ascending brainstem inputs)" or line #334 "It is likely that the observed activity in lateral dorsal POm is driven by true whisker responses in SpVi and S1."

In the same way, Line #315 "we observed POm cells that responded to the onset of the air puff in both conditioning groups". This conclusion should be dampened, to better fit the results, by "we observed POm cells that responded 40 ms after the onset of the air puff in both conditioning groups."

---

## [Referee Report · Reviewer #3 (Public review)]

Petty and Bruno ask whether activity in secondary thalamic nuclei depends on the behavioral relevance of stimulus modality. They recorded from POm and LP, but the weight of the paper is skewed toward POm. They use two cohorts of mice (N=11 and 12), recorded in both nuclei using multi-electrode arrays, while being trained to lick to either a tactile stimulus (air puff against whiskers, first cohort) or a visual stimulus (drifting grating, second cohort), and ignore the respective other. They find that both nuclei, while primarily responsive to their 'home' modality, are more responsive to the relevant modality (i.e. the modality predicting reward).

Strengths:

The paper asks an important question, it is timely and is very well executed. The behavioral method using a delayed lick index (excluding impulsive responses) is well worked out. Electrophysiology methods are state-of-the-art with information about spike quality in Fig. S1. The main result is novel and important, convincingly conveying the point that encoding of secondary thalamic nuclei is flexible and clearly includes aspects of the behavioral relevance of a stimulus. The paper explores the mapping of responses within POm, pointing to a complex functional structure, something that has been reported/suggested in earlier studies.

Weaknesses:

Coding: It does not become clear to which aspect of the task POm/LP are responding. There is a motor-related response (whisking, licking, pupil), which, however, after regressing it out leaves a remaining response that the authors speculate could be sensory.

Learning: The paper talks a lot about 'learning', although it is only indirectly addressed. The authors use two differently (over-)trained mice cohorts rather than studying e.g. a rule switch in one and the same mouse, which would allow to directly assess whether it is the same neurons that undergo rule-dependent encoding

Mapping: The authors present electrode tracks with marked selectivity indices of recordings in POm and LP. This is a great start, but to finally understand the functional composition of POm and LP, a more detailed and systematic mapping effort is needed in the future.

---

## [Author Response]

The following is the authors’ response to the original reviews.

**Reviewer #1 (Public Review):**
Petty and Bruno investigate how response characteristics in the higher-order thalamic nuclei POm (typically somatosensory) and LP (typically visual) change when a stimulus (whisker air puff or visual drifting grating) of one or the other modality is conditioned to a reward. Using a two-step training procedure, they developed an elegant paradigm, where the distractor stimulus is completely uninformative about the reward, which is reflected in the licking behavior of trained mice. While the animals seem to take on to the tactile stimulus more readily, they can also associate the reward with the visual stimulus, ignoring tactile stimuli. In trained mice, the authors recorded single-unit responses in both POm and LP while presenting the same stimuli. The authors first focused on POm recordings, finding that in animals with tactile conditioning POm units specifically responded to the air puff stimulus but not the visual grating. Unexpectedly, in visually conditioned animals, POm units also responded to the visual grating, suggesting that the responses are not modality-specific but more related to behavioral relevance. These effects seem not be homogeneously distributed across POm, whereas lateral units maintain tactile specificity and medial units respond more flexibly. The authors further ask if the unexpected cross-modal responses might result from behavioral activity signatures. By regressing behavior-coupled activity out of the responses, they show that late activity indeed can be related to whisking, licking, and pupil size measures. However, cross-modal short latency responses are not clearly related to animal behavior. Finally, LP neurons also seem to change their modality-specificity dependent on conditioning, whereas tactile responses are attenuated in LP if the animal is conditioned to visual stimuli.The authors make a compelling case that POm neurons are less modality-specific than typically assumed. The training paradigm, employed methods, and analyses are mostly to the point, well supporting the conclusions. The findings importantly widen our understanding of higher-order thalamus processing features with the flexibility to encode multiple modalities and behavioral relevance. The results raise many important questions on the brain-wide representation of conditioned stimuli. E.g. how specific are the responses to the conditioned stimuli? Are thalamic cross-modal neurons recruited for the specific conditioned stimulus or do their responses reflect a more global shift of attention from one modality to another?To elaborate on higher-order thalamic activity in relationship to conditioned behavior, a trialby-trial analysis would be very useful. Is neuronal activity predictive of licking and at which relative timing?

To elaborate on the relationship between neuronal activity and licking, we have created a new supplementary figure (Figure S1), where we present the lick latency of each mouse on the day of recording. We also perform more in-depth analysis of neural activity that occurs before lick onset, which is presented in a new main figure (new Figure 4).

Furthermore, I wonder why the (in my mind) major and from the data obvious take-away, "POm neurons respond more strongly to visual stimuli if visually conditioned", is not directly tested in the summary statistics in Figure 3h.

We have added a summary statistic to Figure 3h and to the Results section (lines 156-157) comparing the drifting grating responses in visually and tactilely conditioned mice.

The remaining early visual responses in POm in visually conditioned mice after removing behavior-linked activity are very convincing (Figure 5d). It would help, however, to see a representation of this on a single-neuron basis side-by-side. Are individual neurons just coupled to behavior while others are independent, or is behaviorally coupled activity a homogeneous effect on all neurons on top of sensory activity?

In lieu of a new figure, we have performed a new analysis of individual neurons to classify them as “stimulus tuned” and/or “movement tuned.” We find that nearly all POm cells encode movement and arousal regardless of whether they also respond to stimuli. This is presented in the Results under the heading “POm correlates with arousal and movement regardless of conditioning” (Lines 219-231).

The conclusions on flexible response characteristics in LP in general are less strongly supported than those in POm. First, the differentiation between POm and LP relies heavily on the histological alignment of labeled probe depth and recording channel, possibly allowing for wrong assignment.

We appreciate the importance in differentiating between POm, LP, and surrounding regions to accurately assign a putative cell to a brain region. The method we employed (aligning an electrode track to a common reference atlas) is widely used in rodent neuroscience, especially in regions like POm and LP which are difficult to differentiate molecularly (for example, see Sibille, Nature Communications, 2022; and Schröder, Neuron, 2020).

Furthermore, it seems surprising, but is not discussed, that putative LP neurons have such strong responses to the air puff stimuli, in both conditioning cases. In tactile conditioning, LP air puff responses seem to be even faster and stronger than POm. In visual conditioning, drifting grating responses paradoxically seem to be later than in tactile conditioning (Fig S2e). These differences in response changes between POm and LP should be discussed in more detail and statements of "similar phenomena" in POm and LP (abstract) should be qualified.

We have further developed our analysis and discussion of LP activity. Our analysis of LP stimulus response latencies are now presented in greater detail in Figure S3, and we have expanded the results section accordingly (lines 266-275). We have also expanded the discussion section to both address these new analyses and speculate on what might drive these surprising “tactile responses” in LP.

**Reviewer #2 (Public Review):**
SummaryThis manuscript by Petty and Bruno delves into the still poorly understood role of higherorder thalamic nuclei in the encoding of sensory information by examining the activity in the Pom and LP cells in mice performing an associative learning task. They developed an elegant paradigm in which they conditioned head-fixed mice to attend to a stimulus of one sensory modality (visual or tactile) and ignore a second stimulus of the other modality. They recorded simultaneously from POm and LP, using 64-channel electrode arrays, to reveal the contextdependency of the firing activity of cells in higher-order thalamic nuclei. They concluded that behavioral training reshapes activity in these secondary thalamic nuclei. I have no major concerns with the manuscript's conclusions, but some important methodological details are lacking and I feel the manuscript could be improved with the following revisions.StrengthsThe authors developed an original and elegant paradigm in which they conditioned headfixed mice to attend to a stimulus of one sensory modality, either visual or tactile, and ignore a second stimulus of the other modality. As a tactile stimulus, they applied gentle air puffs on the distal part of the vibrissae, ensuring that the stimulus was innocuous and therefore none aversive which is crucial in their study.It is commonly viewed that the first-order thalamus performs filtering and re-encoding of the sensory flow; in contrast, the computations taking place in high-order nuclei are poorly understood. They may contribute to cognitive functions. By integrating top-down control, high-order nuclei may participate in generating updated models of the environment based on sensory activity; how this can take place is a key question that Petty and Bruno addressed in the present study.Weaknesses(1) Overall, methods, results, and discussion, involving sensory responses, especially for the Pom, are confusing. I have the feeling that throughout the manuscript, the authors are dealing with the sensory and non-sensory aspects of the modulation of the firing activity in the Pom and LP, without a clear definition of what they examined. Making subsections in the results, or a better naming of what is analyzed could convey the authors' message in a clearer way, e.g., baseline, stim-on, reward.

We thank Reviewer 2 for this suggestion. We have adjusted the language throughout the paper to more clearly state which portions of a given trial we analyzed. We now consistently refer to “baseline,” “stimulus onset,” and “stimulus offset” periods.

In line #502 in Methods, the authors defined "Sensory Responses. We examined each cell's putative sensory response by comparing its firing rate during a "stimulus period" to its baseline firing rate. We first excluded overlapping stimuli, defined as any stimulus occurring within 6 seconds of a stimulus of a different type. We then counted the number of spikes that occurred within 1 second prior to the onset of each stimulus (baseline period) and within one second of the stimulus onset (stimulus period). The period within +/-50ms of the stimulus was considered ambiguous and excluded from analysis."Considering that the responses to whisker deflection, while weak and delayed, were shown to occur, when present, before 50 ms in the Pom (Diamond et al., 1992), it is not clear what the authors mean and consider as "Sensory Responses"?

We have addressed this important concern in three ways. First, we have reanalyzed our data to include the 50ms pre- and post-stimulus time windows that were previously excluded. This did not qualitatively change our results, but updated statistical measurements are reflected in the Results and the legends of figures 3 and 7. Second, we have created a new figure (new Figure 4) which provides a more detailed analysis of early POm stimulus responses at a finer time scale. Third, we have amended the language throughout the paper to refer to “stimulus responses” rather than “sensory responses” to reflect how we cannot disambiguate between bottom-up sensory input and top-down input into POm and LP with our experimental setup. We refer only to “putative sensory responses” when discussing lowlatency (<100ms) stimulus responses.

Precise wording may help to clarify the message. For instance, line #134: "Of cells from tactilely conditioned mice, 175 (50.4%) significantly responded to the air puff, as defined by having a firing rate significantly different from baseline within one second from air puff onset (Figure 3d, bottom)", could be written "significantly responded to the air puff" should be written "significantly increased (or modified if some decreased) their firing rate within one second after the air puff onset (baseline: ...)". This will avoid any confusion with the sensory responses per se.

We have made this specific change suggested by the reviewer (lines 145-146) and made similar adjustments to the language throughout the manuscript to better communicate our analysis methods.

(2) To extend the previous concern, the latency of the modulation of the firing rate of the Pom cells for each modality and each conditioning may be an issue. This latency, given in Figure S2, is rather long, i.e. particularly late latencies for the whisker system, which is completely in favor of non-sensory "responses" per se and the authors' hypothesis that sensory-, arousal-, and movement-evoked activity in Pom are shaped by associative learning. Latency is a key point in this study.Therefore,- latencies should be given in the main text, and Figure S2 could be considered for a main figure, at least panels c, d, and e, could be part of Figure 3.- the Figure S2b points out rather short latency responses to the air puff, at least in some cells, in addition to late ones. The manuscript would highly benefit from an analysis of both early and late latency components of the "responses" to air puffs and drafting grating in both conditions. This analysis may definitely help to clarify the authors' message. Since the authors performed unit recordings, these data are accessible.- it would be highly instructive to examine the latency of the modulation of Pom cells firing rate in parallel with the onset of each behavior, i.e. modification of pupil radius, whisking amplitude, lick rate (Figures 1e, g and 3a, b). The Figure 1 does not provide the latency of the licks in conditioned mice.- the authors mention in the discussion low-latency responses, e.g., line #299: "In both tactilely and visually conditioned mice, movement could not explain the increased firing rate at air puff onset. These low-latency responses across conditioning groups is likely due in part to "true" sensory responses driven by S1 and SpVi."; line #306: "Like POm, LP displayed varied stimulus-evoked activity that was heavily dependent on conditioning. LP responded to the air puff robustly and with low latency, despite lacking direct somatosensory inputs." But which low-latency responses do the authors refer to? Again, this points out that a robust analysis of these latencies is missing in the manuscript but would be helpful to conclude.

We have moved our analysis of stimulus response latency in POm to new Figure 4 in the main text and have expanded both the Results and Discussion sections accordingly. We have also analyzed the lick latency on the day of recording, included in a new supplemental Figure S1.

(3) Anatomical locations of recordings in the dorsal part of the thalamus. Line #122 "Our recordings covered most of the volume of POm but were clustered primarily in the anterior and medial portions of LP (Figure 2d-f). Cells that were within 50 µm of a region border were excluded from analysis."How did the authors distinguish the anterior boundary of the LP with the LD nucleus just more anterior to the LP, another higher-order nucleus, where whisker-responsive cells have been isolated (Bezdudnaya and Keller, 2008)?

Cells within 50µm of any region boundary were excluded, including those at the border of LP and LD. We also reviewed our histology images by eye and believe that our recordings were all made posterior of LD.

(4) The mention in the Methods about the approval by an ethics committee is missing. All the surgery (line #381), i.e., for the implant, the craniotomy, as well as the perfusion, are performed under isoflurane. But isoflurane induces narcosis only and not proper anesthesia. The mention of the use of analgesia is missing.

We thank Reviewer 2 for drawing our attention to this oversight. All experiments were conducted under the approval of the Columbia University IACUC. Mice were treated with the global analgesics buprenorphine and carprofen, the local analgesic bupivacaine, and anesthetized with isoflurane during all surgical procedures. We have amended the Methods section to include this information (Lines 458-470).

**Reviewer #3 (Public Review):**
Petty and Bruno ask whether activity in secondary thalamic nuclei depends on the behavioral relevance of stimulus modality. They recorded from POm and LP, but the weight of the paper is skewed toward POm. They use two cohorts of mice (N=11 and 12), recorded in both nuclei using multi-electrode arrays, while being trained to lick to either a tactile stimulus (air puff against whiskers, first cohort) or a visual stimulus (drifting grating, second cohort), and ignore the respective other. They find that both nuclei, while primarily responsive to their 'home' modality, are more responsive to the relevant modality (i.e. the modality predicting reward).Strengths:The paper asks an important question, it is timely and is very well executed. The behavioral method using a delayed lick index (excluding impulsive responses) is well worked out. Electrophysiology methods are state-of-the-art with information about spike quality in Figure S1. The main result is novel and important, convincingly conveying the point that encoding of secondary thalamic nuclei is flexible and clearly includes aspects of the behavioral relevance of a stimulus. The paper explores the mapping of responses within POm, pointing to a complex functional structure, something that has been reported/suggested in earlier studies.Weaknesses:Coding: It does not become clear to which aspect of the task POm/LP is responding. There is a motor-related response (whisking, licking, pupil), which, however, after regressing it out leaves a remaining response that the authors speculate could be sensory.Learning: The paper talks a lot about 'learning', although it is only indirectly addressed. The authors use two differently (over-)trained mice cohorts rather than studying e.g. a rule switch in one and the same mouse, which would allow us to directly assess whether it is the same neurons that undergo rule-dependent encoding.

We disagree that our animals are “overtrained,” as every mouse was fully trained within 13 days. We agree that it would be interesting to study a rule-switch type experiment, but such an experiment is not necessary to reveal the profound effect that conditioning has on stimulus responses in POm and LP.

Mapping: The authors treat and interpret the two nuclei very much in the same vein, although there are clear differences. I would think these differences are mentioned in passing but could be discussed in more depth. Mapping using responses on electrode tracks is done in POm but not LP.

The mapping of LP responses by anatomical location is presented in the supplemental Figure S4 (previously S3). We have expanded our discussion of LP and how it might differ from POm.

**Reviewer #1 (Recommendations For The Authors):**
Minor writing issues:122 ...67 >LP< cells?301 plural "are”

We have fixed these typos.

Figure issues* 3a,b time ticks are misaligned and the grey bar (bottom) seems not to align with the visual/tactile stimulus shadings.* legend to Figure 3b refers to Figure 1c which is a scheme, but if 1g is meant, this mouse does not seem to have a session 12?* 3c,e time ticks slightly misaligned.* 5e misses shading for the relevant box plots, assuming it should be like Figure 3h.

We thank Reviewer 1 for pointing out these errors. We have adjusted Figures 1, 3, and 5 accordingly.

AnalysesI am missing a similar summary statistics for LP as in Figure 3h

We have added a summary box chart of LP stimulus responses (Figure 7g), similar to that of POm in Figure 3. We have also performed similar statistical analyses, the results of which are presented in the legend for Figure 7.

**Reviewer #2 (Recommendations For The Authors):**
More precisions are required for the following points:(1) The mention of the use of analgesia is missing and this is not a minor concern. Even if the recordings are performed 24 hours after the surgery for the craniotomy and screw insertion and several days after the main surgery for the implant, taking into account the pain of the animals during surgeries is crucial first for ethical reasons, and second because it may affect the data, especially in Pom cells: pain during surgery may induce the development of allodynia and/or hyperalgesia phenomenae and Pom responses to sensory stimuli were shown to be more robust in behavioral hyperalgesia (Masri et al., 2009).

We neglected to include details on the analgesics used during surgery and post-operation recovery in our original manuscript. Mice were administered buprenorphine, carprofen, and bupivacaine immediately prior to the head plate surgery and were treated with additional carprofen during recovery. Mice were similarly treated with analgesics for the craniotomy procedure. Mice were carefully observed after craniotomy, and we saw no evidence of pain or discomfort. Furthermore, mice performed the behavior at the same level pre- and postcraniotomy (now presented in Figure 1j), which also indicates that they were not in any pain.

(2) The head-fixed preparation is only poorly described.Line #414: "Prior to conditioning, mice were habituated to head fixation and given ad libitum water in the behavior apparatus for 15-25 minutes."And line #425 "Mice were trained for one session per day, with each session consisting of an equal number of visual stimuli and air puffs. Sessions ranged from 20-60 minutes and about 40-120 of each stimulus. "More details should be given about the head-fixation training protocol. Are 15-25 minutes the session time duration, 60 minutes, or other time duration? How long does it take to get mice well trained to the head fixation, and on which criteria?Line #389: "Mice were then allowed to recover for 24 hours, after which the sealant was removed and recordings were performed. At the end of experiments,"The timeline is not clear: is there one day or several days of recordings?

We have expanded on our description of the head fixation protocol in the Methods. We describe in more detail how mice were habituated to head fixation, the timing of water restriction, and the start of conditioning/training (Habituation and Conditioning, lines 492-500).

(4) Line #411: "Mice were deprived of water 3 days prior to the start of conditioning" followed by line #414 "Prior to conditioning, mice were habituated to head fixation and given ad libitum water in the behavior apparatus for 15-25 minutes".If I understood correctly, the mice were then not fully water-deprived for 3 days since they received water while head-fixed. This point may be clarified.

We addressed these concerns in the changes to the Methods section mentioned in the preceding point (3).

(5) Line #157: "Modality selectivity varies with anatomical location in Pom" while the end of the previous paragraph is "This suggests that POm encoding of reward and/or licking is insensitive to task type, an observation we examine further below."The authors then come to anatomical concerns before coming back to what the Pom may encode in the following section. This makes the story quite confusing and hard to follow even though pretty interesting.

We have reordered our Figures and Results to improve the flow of the paper and remove this point of confusion. We now present results on the encoding of movement before analyzing the relationship between POm stimulus responses and anatomical location. What was old Figure 5 now precedes what was old Figure 4.

(6) Licks Analysis. Line #99 "However, this mouse also learned that the air puff predicted a lack of reward in the shaping task, as evidenced by withholding licking upon the onset of the air puff. The mouse thus displayed a positive visual lick index and a negative tactile lick index, suggesting that it attended to both the tactile and visual stimuli (Figure 1f, middle arrow)."Line #105 "All visually conditioned mice exhibited a similar learning trajectory (Figure 1i left, 1j left)".Interestingly, the authors revealed that mice withheld licking upon the onset of the air puff in the visual conditioning, which they did not do at the onset of the drifting grating in the tactile conditioning. This withholding was extinguished after the 8th session, which the authors interpret as the mice finally ignoring the air puff. Is this effect significant, is there a significant withholding licking upon the onset of the air puff on the 12 tested mice?

The withholding of licking was significant (assessed with a sign-rank test) in visually conditioned mice prior to switching to the full version of the task. Indeed, it was the abolishment of this effect after conditioning with the full version of the task that was our criterion for when a mouse was fully trained. We have elaborated on this in the Habituation and Conditioning section in the Methods.

(1) Throughout the manuscript "Touch" is used instead of passive whisker deflection, and may be confusing with "active touch" for the whisker community readers. I recommend avoiding using "touch" instead of "passive whisker deflection".

We appreciate that “touch” can be an ambiguous term in some contexts. However, we have limited our use of the word to refer to the percept of whisker deflection; we do not describe the air puff stimulus as a “touch.” We respectfully would like to retain the use of the word, as it is useful for comparing somatosensory stimuli to visual stimuli.

(2) Line #395: "Air puffs (0.5-1 PSI) were delivered through a nozzle (cut p1000 pipet tip, approximately 3.5mm diameter aperture)".Are air puffs of <1 PSI applied, not <1 bar?

We thank Reviewer 3 for pointing out this inaccuracy. The air puffs were indeed between 0.5 and 1 bar, not PSI. We have addressed this in the Methods.

(3) Line #441: "In the full task, the stimuli and reward were identical, but stimuli were presented at uncorrelated and less predictable intervals." Do the authors mean that all stimuli are rewarded?

The stimuli and reward were identical between the shaping and full versions of the task. In the full version of the task, the unrewarded stimulus was truly uncorrelated with reward, rather than anticorrelated.

(4) Line #445 "for a mean ISI of 20 msec." ISI is not defined, I guess that it means interstimulus interval. Even if pretty obvious, to avoid any confusion for future readers, I would recommend using another acronym, especially in a manuscript about electrophysiology, since ISI is a dedicated acronym for inter-spike interval.

We have defined the acronym ISI as “inter-stimulus interval” when first introduced in the results (Line 82) and in the Methods (Line 511).

(5) Line #416 "In the first phase of conditioning ("shaping"), mice were separated into two cohorts: a "tactile" cohort and a "visual" cohort. Mice were presented with tactile stimuli (a two-second air puff delivered to the distal whisker field) and visual stimuli (vertical drifting grating on a monitor). Throughout conditioning, mice were monitored via webcam to ensure that the air puff only contacted the whiskers and did not disturb the facial fur nor cause the mouse to blink, flinch, or otherwise react - ensuring the stimulus was innocuous. The stimulus types were randomly ordered. In the visual conditioning cohort, the visual stimulus was paired with a water reward (8-16µL) delivered at the time of stimulus offset. In the tactile conditioning cohort, the reward was instead paired with the offset of the air puff. Regardless of the type of conditioning, stimulus type was a balanced 50:50 with an inter-stimulus interval of 8-12 seconds (uniform distribution)."The mention of the "full version of the task" will be welcome in this paragraph to clarify what the task is for the mouse in the Methods part.

We have more clearly defined the full version of the task in a later paragraph (line 506). We believe this addresses the potential confusion caused by the original description of the conditioning paradigm.

(6) Line #467: "Units were assigned to the array channel on which its mean waveform was largest".Should it read mean waveform "amplitude"?

This is correct, we have adjusted the statement accordingly.

(7) Line #482 "The eye camera was positioned on the right side of the face and recorded at 60 fps." Then line #487 "The trace of pupil radius over time was smoothed over 5 frames (8.3 msec).” 5 frames, with a 60fps, represent then 83 ms and not 8.3 ms.

We have corrected this error.

(8) Line #121: "257 POm cells and 67 cells from 12 visually conditioned mice"67 LP cells, LP is missing

We have corrected this error.

(9) Line #354: "A consistent result of attention studies in humans and nonhuman primates is the enhancement of cortical and thalamic sensory responses to an attended visual stimuli. Here, we show not just enhancement of sensory responses to stimuli within a single modality, but also across modalities. It is worth investigating further how secondary thalamus and high-order sensory cortex encode attention to stimuli outside of their respective modalities. Our surprising conclusion that the nuclei are equivalently activated by behaviorally relevant stimuli is nevertheless compatible with these previous studies." Since higher-order thalamic nuclei are integrative centers of many cortical and subcortical inputs, they cannot be viewed simply as relay nuclei, and there is therefore no "surprising" conclusion in these results. Not surprising, but still an elegant demonstration of the contextdependent activity/responses of the Pom/LP cells.

We disagree. Visual stimuli activating strong POm responses and tactile stimuli activating strong LP responses - however they do it - is a surprising result. We agree that higher-order thalamic nuclei are integrative centers, but exactly what they integrate and what the integrated output means is still poorly understood.